# Policy Optimization via Importance Sampling

**Alberto Maria Metelli**
Politecnico di Milano, Milan, Italy
`albertomaria.metelli@polimi.it`

**Matteo Papini**
Politecnico di Milano, Milan, Italy
`matteo.papini@polimi.it`

**Francesco Faccio**
Politecnico di Milano, Milan, Italy
IDSIA, USI-SUPSI, Lugano, Switzerland
`francesco.faccio@mail.polimi.it`

**Marcello Restelli**
Politecnico di Milano, Milan, Italy
`marcello.restelli@polimi.it`

## Abstract

Policy optimization is an effective reinforcement learning approach to solve continuous control tasks. Recent achievements have shown that alternating online and offline optimization is a successful choice for efficient trajectory reuse. However, deciding when to stop optimizing and collect new trajectories is non-trivial, as it requires to account for the variance of the objective function estimate. In this paper, we propose a novel, model-free, policy search algorithm, POIS, applicable in both action-based and parameter-based settings. We first derive a high-confidence bound for importance sampling estimation; then we define a surrogate objective function, which is optimized offline whenever a new batch of trajectories is collected. Finally, the algorithm is tested on a selection of continuous control tasks, with both linear and deep policies, and compared with state-of-the-art policy optimization methods.

## 1 Introduction

In recent years, policy search methods [10] have proved to be valuable Reinforcement Learning (RL) [50] approaches thanks to their successful achievements in continuous control tasks [e.g., 23, 42, 44, 43], robotic locomotion [e.g., 53, 20] and partially observable environments [e.g., 28]. These algorithms can be roughly classified into two categories: *action-based* methods [51, 34] and *parameter-based* methods [45]. The former, usually known as policy gradient (PG) methods, perform a search in a parametric policy space by following the gradient of the utility function estimated by means of a batch of trajectories collected from the environment [50]. In contrast, in parameter-based methods, the search is carried out directly in the space of parameters by exploiting global optimizers [e.g., 41, 16, 48, 52] or following a proper gradient direction like in Policy Gradients with Parameter-based Exploration (PGPE) [45, 63, 46]. A major question in policy search methods is: how should we use a batch of trajectories in order to exploit its information in the most efficient way? On one hand, *on-policy* methods leverage on the batch to perform a single gradient step, after which new trajectories are collected with the updated policy. Online PG methods are likely the most widespread policy search approaches: starting from the traditional algorithms based on stochastic policy gradient [51], like REINFORCE [64] and G(PO)MDP [4], moving toward more modern methods, such as Trust Region Policy Optimization (TRPO) [42]. These methods, however, rarely exploit the available trajectories in an efficient way, since each batch is thrown away after just one gradient update. On the other hand, *off-policy* methods maintain a behavioral policy, used to explore the environment and to collect samples, and a target policy which is optimized. The concept of off-policy learning is rooted in value-based RL [62, 30, 27] and it was first adapted to PG in [9], using an actor-critic architecture. The approach has been extended to Deterministic Policy Gradient (DPG) [47], which allows optimizing deterministic policies while keeping a stochastic policy for exploration.

More recently, an efficient version of DPG coupled with a deep neural network to represent the policy has been proposed, named Deep Deterministic Policy Gradient (DDPG) [23]. In the parameter-based framework, even though the original formulation [45] introduces an online algorithm, an extension has been proposed to efficiently reuse the trajectories in an offline scenario [67]. Furthermore, PGPE-like approaches allow overcoming several limitations of classical PG, like the need for a stochastic policy and the high variance of the gradient estimates.[1]

While on-policy algorithms are, by nature, *online*, as they need to be fed with fresh samples whenever the policy is updated, off-policy methods can take advantage of mixing online and *offline* optimization. This can be done by alternately sampling trajectories and performing optimization epochs with the collected data. A prime example of this alternating procedure is Proximal Policy Optimization (PPO) [44], that has displayed remarkable performance on continuous control tasks. Off-line optimization, however, introduces further sources of approximation, as the gradient w.r.t. the target policy needs to be estimated (off-policy) with samples collected with a behavioral policy. A common choice is to adopt an *importance sampling* (IS) [29, 17] estimator in which each sample is reweighted proportionally to the likelihood of being generated by the target policy. However, directly optimizing this utility function is impractical since it displays a wide variance most of the times [29]. Intuitively, the variance increases proportionally to the distance between the behavioral and the target policy; thus, the estimate is reliable as long as the two policies are close enough. Preventing uncontrolled updates in the space of policy parameters is at the core of the natural gradient approaches [1] applied effectively both on PG methods [18, 33, 63] and on PGPE methods [26]. More recently, this idea has been captured (albeit indirectly) by TRPO, which optimizes via (approximate) natural gradient a surrogate objective function, derived from safe RL [18, 35], subject to a constraint on the Kullback-Leibler divergence between the behavioral and target policy.[2] Similarly, PPO performs a truncation of the importance weights to discourage the optimization process from going too far. Although TRPO and PPO, together with DDPG, represent the state-of-the-art policy optimization methods in RL for continuous control, they do not explicitly encode in their objective function the uncertainty injected by the importance sampling procedure. A more theoretically grounded analysis has been provided for policy selection [11], model-free [56] and model-based [54] policy evaluation (also accounting for samples collected with multiple behavioral policies), and combined with options [15]. Subsequently, in [55] these methods have been extended for policy improvement, deriving a suitable concentration inequality for the case of truncated importance weights. Unfortunately, these methods are hardly scalable to complex control tasks. A more detailed review of the state-of-the-art policy optimization algorithms is reported in Appendix A.

In this paper, we propose a novel, model-free, actor-only, policy optimization algorithm, named *Policy Optimization via Importance Sampling* (POIS) that mixes online and offline optimization to efficiently exploit the information contained in the collected trajectories. POIS explicitly accounts for the uncertainty introduced by the importance sampling by optimizing a surrogate objective function. The latter captures the trade-off between the estimated performance improvement and the variance injected by the importance sampling. The main contributions of this paper are theoretical, algorithmic and experimental. After revising some notions about importance sampling (Section 3), we propose a concentration inequality, of independent interest, for high-confidence "off-distribution" optimization of objective functions estimated via importance sampling (Section 4). Then we show how this bound can be customized into a surrogate objective function in order to either search in the space of policies (Action-based POIS) or to search in the space of parameters (Parameter-bases POIS). The resulting algorithm (in both the action-based and the parameter-based flavor) collects, at each iteration, a set of trajectories. These are used to perform offline optimization of the surrogate objective via gradient ascent (Section 5), after which a new batch of trajectories is collected using the optimized policy. Finally, we provide an experimental evaluation with both linear policies and deep neural policies to illustrate the advantages and limitations of our approach compared to state-of-the-art algorithms (Section 6) on classical control tasks [12, 57]. The proofs for all Theorems and Lemmas can be found in Appendix B. The implementation of POIS can be found at `https://github.com/T3p/pois`.

## 2 Preliminaries

A discrete-time Markov Decision Process (MDP) [37] is defined as a tuple $\mathcal{M} = (\mathcal{S}, \mathcal{A}, P, R, \gamma, D)$ where $\mathcal{S}$ is the state space, $\mathcal{A}$ is the action space, $P(\cdot|s, a)$ is a Markovian transition model that assigns for each state-action pair $(s, a)$ the probability of reaching the next state $s'$, $\gamma \in [0, 1]$ is the discount factor, $R(s, a) \in [-R_{\max}, R_{\max}]$ assigns the expected reward for performing action $a$ in state $s$ and $D$ is the distribution of the initial state. The behavior of an agent is described by a policy $\pi(\cdot|s)$ that assigns for each state $s$ the probability of performing action $a$. A trajectory $\tau \in \mathcal{T}$ is a sequence of state-action pairs $\tau = (s_{\tau,0}, a_{\tau,0}, \ldots, s_{\tau,H-1}, a_{\tau,H-1}, s_{\tau,H})$, where $H$ is the actual trajectory horizon. The performance of an agent is evaluated in terms of the *expected return*, i.e., the expected discounted sum of the rewards collected along the trajectory: $\mathbb{E}_\tau [R(\tau)]$, where $R(\tau) = \sum_{t=0}^{H-1} \gamma^t R(s_{\tau,t}, a_{\tau,t})$ is the trajectory return.

We focus our attention to the case in which the policy belongs to a parametric policy space $\Pi_\Theta = \{\pi_{\boldsymbol{\theta}} : \boldsymbol{\theta} \in \Theta \subseteq \mathbb{R}^p\}$. In parameter-based approaches, the agent is equipped with a *hyperpolicy* $\nu$ used to sample the policy parameters at the beginning of each episode. The hyperpolicy belongs itself to a parametric hyperpolicy space $\mathcal{N}_\mathcal{P} = \{\nu_{\boldsymbol{\rho}} : \boldsymbol{\rho} \in \mathcal{P} \subseteq \mathbb{R}^r\}$. The expected return can be expressed, in the parameter-based case, as a double expectation: one over the policy parameter space $\Theta$ and one over the trajectory space $\mathcal{T}$:

$$J_D(\boldsymbol{\rho}) = \int_\Theta \int_\mathcal{T} \nu_{\boldsymbol{\rho}}(\boldsymbol{\theta}) p(\tau|\boldsymbol{\theta}) R(\tau) \, \mathrm{d}\tau \, \mathrm{d}\boldsymbol{\theta}, \tag{1}$$

where $p(\tau|\boldsymbol{\theta}) = D(s_0) \prod_{t=0}^{H-1} \pi_{\boldsymbol{\theta}}(a_t|s_t) P(s_{t+1}|s_t, a_t)$ is the trajectory density function. The goal of a parameter-based learning agent is to determine the hyperparameters $\boldsymbol{\rho}^*$ so as to maximize $J_D(\boldsymbol{\rho})$. If $\nu_{\boldsymbol{\rho}}$ is stochastic and differentiable, the hyperparameters can be learned according to the gradient ascent update: $\boldsymbol{\rho}' = \boldsymbol{\rho} + \alpha \nabla_{\boldsymbol{\rho}} J_D(\boldsymbol{\rho})$, where $\alpha > 0$ is the step size and $\nabla_{\boldsymbol{\rho}} J_D(\boldsymbol{\rho}) = \int_\Theta \int_\mathcal{T} \nu_{\boldsymbol{\rho}}(\boldsymbol{\theta}) p(\tau|\boldsymbol{\theta}) \nabla_{\boldsymbol{\rho}} \log \nu_{\boldsymbol{\rho}}(\boldsymbol{\theta}) R(\tau) \, \mathrm{d}\tau \, \mathrm{d}\boldsymbol{\theta}$. Since the stochasticity of the hyperpolicy is a sufficient source of exploration, deterministic action policies of the kind $\pi_{\boldsymbol{\theta}}(a|s) = \delta(a - u_{\boldsymbol{\theta}}(s))$ are typically considered, where $\delta$ is the Dirac delta function and $u_{\boldsymbol{\theta}}$ is a deterministic mapping from $\mathcal{S}$ to $\mathcal{A}$. In the action-based case, on the contrary, the hyperpolicy $\nu_{\boldsymbol{\rho}}$ is a deterministic distribution $\nu_{\boldsymbol{\rho}}(\boldsymbol{\theta}) = \delta(\boldsymbol{\theta} - g(\boldsymbol{\rho}))$, where $g(\boldsymbol{\rho})$ is a deterministic mapping from $\mathcal{P}$ to $\Theta$. For this reason, the dependence on $\boldsymbol{\rho}$ is typically not represented and the expected return expression simplifies into a single expectation over the trajectory space $\mathcal{T}$:

$$J_D(\boldsymbol{\theta}) = \int_\mathcal{T} p(\tau|\boldsymbol{\theta}) R(\tau) \, \mathrm{d}\tau. \tag{2}$$

An action-based learning agent aims to find the policy parameters $\boldsymbol{\theta}^*$ that maximize $J_D(\boldsymbol{\theta})$. In this case, we need to enforce exploration by means of the stochasticity of $\pi_{\boldsymbol{\theta}}$. For stochastic and differentiable policies, learning can be performed via gradient ascent: $\boldsymbol{\theta}' = \boldsymbol{\theta} + \alpha \nabla_{\boldsymbol{\theta}} J_D(\boldsymbol{\theta})$, where $\nabla_{\boldsymbol{\theta}} J_D(\boldsymbol{\theta}) = \int_\mathcal{T} p(\tau|\boldsymbol{\theta}) \nabla_{\boldsymbol{\theta}} \log p(\tau|\boldsymbol{\theta}) R(\tau) \, \mathrm{d}\tau$.

## 3 Evaluation via Importance Sampling

In off-policy evaluation [56, 54], we aim to estimate the performance of a target policy $\pi_T$ (or hyperpolicy $\nu_T$) given samples collected with a behavioral policy $\pi_B$ (or hyperpolicy $\nu_B$). More generally, we face the problem of estimating the expected value of a deterministic bounded function $f$ ($\|f\|_\infty < +\infty$) of random variable $x$ taking values in $\mathcal{X}$ under a target distribution $P$, after having collected samples from a behavioral distribution $Q$. The *importance sampling* estimator (IS) [7, 29] corrects the distribution with the *importance weights* (or Radon–Nikodym derivative or likelihood ratio) $w_{P/Q}(x) = p(x)/q(x)$:

$$\widehat{\mu}_{P/Q} = \frac{1}{N} \sum_{i=1}^N \frac{p(x_i)}{q(x_i)} f(x_i) = \frac{1}{N} \sum_{i=1}^N w_{P/Q}(x_i) f(x_i), \tag{3}$$

where $\mathbf{x} = (x_1, x_2, \ldots, x_N)^T$ is sampled from $Q$ and we assume $q(x) > 0$ whenever $f(x)p(x) \neq 0$. This estimator is unbiased ($\mathbb{E}_{\mathbf{x} \sim Q}[\widehat{\mu}_{P/Q}] = \mathbb{E}_{x \sim P}[f(x)]$) but it may exhibit an undesirable behavior due to the variability of the importance weights, showing, in some cases, infinite variance. Intuitively, the magnitude of the importance weights provides an indication of how much the probability measures $P$ and $Q$ are dissimilar. This notion can be formalized by the Rényi divergence [40, 59], an information-theoretic dissimilarity index between probability measures.

**Rényi divergence** Let $P$ and $Q$ be two probability measures on a measurable space $(\mathcal{X}, \mathcal{F})$ such that $P \ll Q$ ($P$ is absolutely continuous w.r.t. $Q$) and $Q$ is $\sigma$-finite. Let $P$ and $Q$ admit $p$ and $q$ as Lebesgue probability density functions (p.d.f.), respectively. The $\alpha$-Rényi divergence is defined as:

$$D_\alpha(P\|Q) = \frac{1}{\alpha - 1} \log \int_{\mathcal{X}} \left( \frac{\mathrm{d}P}{\mathrm{d}Q} \right)^\alpha \mathrm{d}Q = \frac{1}{\alpha - 1} \log \int_{\mathcal{X}} q(x) \left( \frac{p(x)}{q(x)} \right)^\alpha \mathrm{d}x, \qquad (4)$$

where $\mathrm{d}P/\mathrm{d}Q$ is the Radon–Nikodym derivative of $P$ w.r.t. $Q$ and $\alpha \in [0, \infty]$. Some remarkable cases are: $\alpha = 1$ when $D_1(P\|Q) = D_{\mathrm{KL}}(P\|Q)$ and $\alpha = \infty$ yielding $D_\infty(P\|Q) = \log \mathrm{ess\,sup}_{\mathcal{X}} \mathrm{d}P/\mathrm{d}Q$. Importing the notation from [8], we indicate the exponentiated $\alpha$-Rényi divergence as $d_\alpha(P\|Q) = \exp(D_\alpha(P\|Q))$. With little abuse of notation, we will replace $D_\alpha(P\|Q)$ with $D_\alpha(p\|q)$ whenever possible within the context.

The Rényi divergence provides a convenient expression for the moments of the importance weights: $\mathbb{E}_{x \sim Q}\left[ w_{P/Q}(x)^\alpha \right] = d_\alpha(P\|Q)$. Moreover, $\mathbb{V}\mathrm{ar}_{x \sim Q}\left[ w_{P/Q}(x) \right] = d_2(P\|Q) - 1$ and $\mathrm{ess\,sup}_{x \sim Q} w_{P/Q}(x) = d_\infty(P\|Q)$ [8]. To mitigate the variance problem of the IS estimator, we can resort to the *self-normalized importance sampling* estimator (SN) [7]:

$$\widetilde{\mu}_{P/Q} = \frac{\sum_{i=1}^N w_{P/Q}(x_i) f(x_i)}{\sum_{i=1}^N w_{P/Q}(x_i)} = \sum_{i=1}^N \widetilde{w}_{P/Q}(x_i) f(x_i), \qquad (5)$$

where $\widetilde{w}_{P/Q}(x) = w_{P/Q}(x)/\sum_{i=1}^N w_{P/Q}(x_i)$ is the self-normalized importance weight. Differently from $\widehat{\mu}_{P/Q}$, $\widetilde{\mu}_{P/Q}$ is biased but consistent [29] and it typically displays a more desirable behavior because of its smaller variance.[3] Given the realization $x_1, x_2, \ldots, x_N$ we can interpret the SN estimator as the expected value of $f$ under an approximation of the distribution $P$ made by $N$ deltas, i.e., $\widetilde{p}(x) = \sum_{i=1}^N \widetilde{w}_{P/Q}(x) \delta(x - x_i)$. The problem of assessing the quality of the SN estimator has been extensively studied by the simulation community, producing several diagnostic indexes to indicate when the weights might display problematic behavior [29]. The *effective sample size* (ESS) was introduced in [22] as the number of samples drawn from $P$ so that the variance of the Monte Carlo estimator $\widetilde{\mu}_{P/P}$ is approximately equal to the variance of the SN estimator $\widetilde{\mu}_{P/Q}$ computed with $N$ samples. Here we report the original definition and its most common estimate:

$$\mathrm{ESS}(P\|Q) = \frac{N}{\mathbb{V}\mathrm{ar}_{x \sim Q}\left[ w_{P/Q}(x) \right] + 1} = \frac{N}{d_2(P\|Q)}, \quad \widehat{\mathrm{ESS}}(P\|Q) = \frac{1}{\sum_{i=1}^N \widetilde{w}_{P/Q}(x_i)^2}. \qquad (6)$$

The ESS has an interesting interpretation: if $d_2(P\|Q) = 1$, i.e., $P = Q$ almost everywhere, then ESS $= N$ since we are performing Monte Carlo estimation. Otherwise, the ESS decreases as the dissimilarity between the two distributions increases. In the literature, other ESS-like diagnostics have been proposed that also account for the nature of $f$ [24].

## 4 Optimization via Importance Sampling

The off-policy optimization problem [55] can be formulated as finding the best target policy $\pi_T$ (or hyperpolicy $\nu_T$), i.e., the one maximizing the expected return, having access to a set of samples collected with a behavioral policy $\pi_B$ (or hyperpolicy $\nu_B$). In a more abstract sense, we aim to determine the target distribution $P$ that maximizes $\mathbb{E}_{x \sim P}[f(x)]$ having samples collected from the fixed behavioral distribution $Q$. In this section, we analyze the problem of defining a proper objective function for this purpose. Directly optimizing the estimator $\widehat{\mu}_{P/Q}$ or $\widetilde{\mu}_{P/Q}$ is, in most of the cases, unsuccessful. With enough freedom in choosing $P$, the optimal solution would assign as much probability mass as possible to the maximum value among $f(x_i)$. Clearly, in this scenario, the estimator is unreliable and displays a large variance. For this reason, we adopt a risk-averse approach and we decide to optimize a statistical *lower bound* of the expected value $\mathbb{E}_{x \sim P}[f(x)]$ that holds with high confidence. We start by analyzing the behavior of the IS estimator and we provide the following result that bounds the variance of $\widehat{\mu}_{P/Q}$ in terms of the Renyi divergence.

**Lemma 4.1.** *Let $P$ and $Q$ be two probability measures on the measurable space $(\mathcal{X}, \mathcal{F})$ such that $P \ll Q$. Let $\mathbf{x} = (x_1, x_2, \ldots, x_N)^T$ i.i.d. random variables sampled from $Q$ and $f : \mathcal{X} \to \mathbb{R}$ be a*

bounded function ($\|f\|_\infty < +\infty$). Then, for any $N > 0$, the variance of the IS estimator $\widehat{\mu}_{P/Q}$ can be upper bounded as:

$$\underset{\mathbf{x} \sim Q}{\mathbb{V}\text{ar}} \left[ \widehat{\mu}_{P/Q} \right] \leq \frac{1}{N} \|f\|_\infty^2 d_2 \left( P \| Q \right). \tag{7}$$

When $P = Q$ almost everywhere, we get $\mathbb{V}\text{ar}_{\mathbf{x} \sim Q} \left[ \widehat{\mu}_{Q/Q} \right] \leq \frac{1}{N} \|f\|_\infty^2$, a well-known bound on the variance of a Monte Carlo estimator. Recalling the definition of ESS (6) we can rewrite the previous bound as: $\mathbb{V}\text{ar}_{\mathbf{x} \sim Q} \left[ \widehat{\mu}_{P/Q} \right] \leq \frac{\|f\|_\infty^2}{\text{ESS}(P\|Q)}$, i.e., the variance scales with ESS instead of $N$. While $\widehat{\mu}_{P/Q}$ can have unbounded variance even if $f$ is bounded, the SN estimator $\widetilde{\mu}_{P/Q}$ is always bounded by $\|f\|_\infty$ and therefore it always has a finite variance. Since the normalization term makes all the samples $\widetilde{w}_{P/Q}(x_i) f(x_i)$ interdependent, an exact analysis of its bias and variance is more challenging. Several works adopted approximate methods to provide an expression for the variance [17]. We propose an analysis of bias and variance of the SN estimator in Appendix D.

## 4.1 Concentration Inequality

Finding a suitable concentration inequality for off-policy learning was studied in [56] for offline policy evaluation and subsequently in [55] for optimization. On one hand, fully empirical concentration inequalities, like Student-T, besides the asymptotic approximation, are not suitable in this case since the empirical variance needs to be estimated with importance sampling as well injecting further uncertainty [29]. On the other hand, several distribution-free inequalities like Hoeffding require knowing the maximum of the estimator, which might not exist ($d_\infty(P\|Q) = \infty$) for the IS estimator. Constraining $d_\infty(P\|Q)$ to be finite often introduces unacceptable limitations. For instance, in the case of univariate Gaussian distributions, it prevents a step that selects a target variance larger than the behavioral one from being performed (see Appendix C).[4] Even Bernstein inequalities [5], are hardly applicable since, for instance, in the case of univariate Gaussian distributions, the importance weights display a fat tail behavior (see Appendix C). We believe that a reasonable trade-off is to require the variance of the importance weights to be finite, that is equivalent to require $d_2(P\|Q) < \infty$, i.e., $\sigma_P < 2\sigma_Q$ for univariate Gaussians. For this reason, we resort to Chebyshev-like inequalities and we propose the following concentration bound derived from Cantelli's inequality and customized for the IS estimator.

**Theorem 4.1.** *Let $P$ and $Q$ be two probability measures on the measurable space $(\mathcal{X}, \mathcal{F})$ such that $P \ll Q$ and $d_2(P\|Q) < +\infty$. Let $x_1, x_2, \ldots, x_N$ be i.i.d. random variables sampled from $Q$, and $f : \mathcal{X} \to \mathbb{R}$ be a bounded function ($\|f\|_\infty < +\infty$). Then, for any $0 < \delta \leq 1$ and $N > 0$ with probability at least $1 - \delta$ it holds that:*

$$\underset{x \sim P}{\mathbb{E}} \left[ f(x) \right] \geq \frac{1}{N} \sum_{i=1}^{N} w_{P/Q}(x_i) f(x_i) - \|f\|_\infty \sqrt{\frac{(1-\delta) d_2(P\|Q)}{\delta N}}. \tag{8}$$

The bound highlights the interesting trade-off between the estimated performance and the uncertainty introduced by changing the distribution. The latter enters in the bound as the 2-Rényi divergence between the target distribution $P$ and the behavioral distribution $Q$. Intuitively, we should trust the estimator $\widehat{\mu}_{P/Q}$ as long as $P$ is not too far from $Q$. For the SN estimator, accounting for the bias, we are able to obtain a bound (reported in Appendix D), with a similar dependence on $P$ as in Theorem 4.1, albeit with different constants. Renaming all constants involved in the bound of Theorem 4.1 as $\lambda = \|f\|_\infty \sqrt{(1 - \delta)/\delta}$, we get a surrogate objective function. The optimization can be carried out in different ways. The following section shows why using the natural gradient could be a successful choice in case $P$ and $Q$ can be expressed as parametric differentiable distributions.

## 4.2 Importance Sampling and Natural Gradient

We can look at a parametric distribution $P_{\boldsymbol{\omega}}$, having $p_{\boldsymbol{\omega}}$ as a density function, as a point on a probability manifold with coordinates $\boldsymbol{\omega} \in \Omega$. If $p_{\boldsymbol{\omega}}$ is differentiable, the Fisher Information Matrix (FIM) [39, 2] is defined as: $\mathcal{F}(\boldsymbol{\omega}) = \int_\mathcal{X} p_{\boldsymbol{\omega}}(x) \nabla_{\boldsymbol{\omega}} \log p_{\boldsymbol{\omega}}(x) \nabla_{\boldsymbol{\omega}} \log p_{\boldsymbol{\omega}}(x)^T \, \mathrm{d}x$. This matrix is, up to a scale, an

**Algorithm 1** Action-based POIS

Initialize $\boldsymbol{\theta}_0^0$ arbitrarily
**for** $j = 0, 1, 2, ...,$ until convergence **do**
    Collect $N$ trajectories with $\pi_{\boldsymbol{\theta}_0^j}$
    **for** $k = 0, 1, 2, ...,$ until convergence **do**
        Compute $\mathcal{G}(\boldsymbol{\theta}_k^j)$, $\nabla_{\boldsymbol{\theta}_k^j} \mathcal{L}(\boldsymbol{\theta}_k^j / \boldsymbol{\theta}_0^j)$ and $\alpha_k$
        $\boldsymbol{\theta}_{k+1}^j = \boldsymbol{\theta}_k^j + \alpha_k \mathcal{G}(\boldsymbol{\theta}_k^j)^{-1} \nabla_{\boldsymbol{\theta}_k^j} \mathcal{L}(\boldsymbol{\theta}_k^j / \boldsymbol{\theta}_0^j)$
    **end for**
    $\boldsymbol{\theta}_0^{j+1} = \boldsymbol{\theta}_k^j$
**end for**

**Algorithm 2** Parameter-based POIS

Initialize $\boldsymbol{\rho}_0^0$ arbitrarily
**for** $j = 0, 1, 2, ...,$ until convergence **do**
    Sample $N$ policy parameters $\boldsymbol{\theta}_i^j$ from $\nu_{\boldsymbol{\rho}_0^j}$
    Collect a trajectory with each $\pi_{\boldsymbol{\theta}_i^j}$
    **for** $k = 0, 1, 2, ...,$ until convergence **do**
        Compute $\mathcal{G}(\boldsymbol{\rho}_k^j)$, $\nabla_{\boldsymbol{\rho}_k^j} \mathcal{L}(\boldsymbol{\rho}_k^j / \boldsymbol{\rho}_0^j)$ and $\alpha_k$
        $\boldsymbol{\rho}_{k+1}^j = \boldsymbol{\rho}_k^j + \alpha_k \mathcal{G}(\boldsymbol{\rho}_k^j)^{-1} \nabla_{\boldsymbol{\rho}_k^j} \mathcal{L}(\boldsymbol{\rho}_k^j / \boldsymbol{\rho}_0^j)$
    **end for**
    $\boldsymbol{\rho}_0^{j+1} = \boldsymbol{\rho}_k^j$
**end for**

invariant metric [1] on parameter space $\Omega$, i.e., $\kappa(\boldsymbol{\omega}' - \boldsymbol{\omega})^T \mathcal{F}(\boldsymbol{\omega})(\boldsymbol{\omega}' - \boldsymbol{\omega})$ is independent on the specific parameterization and provides a second order approximation of the distance between $p_{\boldsymbol{\omega}}$ and $p_{\boldsymbol{\omega}'}$ on the probability manifold up to a scale factor $\kappa \in \mathbb{R}$. Given a loss function $\mathcal{L}(\boldsymbol{\omega})$, we define the natural gradient [1, 19] as $\widetilde{\nabla}_{\boldsymbol{\omega}} \mathcal{L}(\boldsymbol{\omega}) = \mathcal{F}^{-1}(\boldsymbol{\omega}) \nabla_{\boldsymbol{\omega}} \mathcal{L}(\boldsymbol{\omega})$, which represents the steepest ascent direction in the probability manifold. Thanks to the invariance property, there is a tight connection between the geometry induced by the Rényi divergence and the Fisher information metric [3].

**Theorem 4.2.** *Let $p_{\boldsymbol{\omega}}$ be a p.d.f. differentiable w.r.t. $\boldsymbol{\omega} \in \Omega$. Then, it holds that, for the Rényi divergence: $D_\alpha(p_{\boldsymbol{\omega}'} \| p_{\boldsymbol{\omega}}) = \frac{\alpha}{2} (\boldsymbol{\omega}' - \boldsymbol{\omega})^T \mathcal{F}(\boldsymbol{\omega}) (\boldsymbol{\omega}' - \boldsymbol{\omega}) + o(\|\boldsymbol{\omega}' - \boldsymbol{\omega}\|_2^2)$, and for the exponentiated Rényi divergence: $d_\alpha(p_{\boldsymbol{\omega}'} \| p_{\boldsymbol{\omega}}) = 1 + \frac{\alpha}{2} (\boldsymbol{\omega}' - \boldsymbol{\omega})^T \mathcal{F}(\boldsymbol{\omega}) (\boldsymbol{\omega}' - \boldsymbol{\omega}) + o(\|\boldsymbol{\omega}' - \boldsymbol{\omega}\|_2^2)$.*

This result provides an approximate expression for the variance of the importance weights, as $\mathbb{V}\mathrm{ar}_{x \sim p_{\boldsymbol{\omega}}} \left[ w_{\boldsymbol{\omega}'/\boldsymbol{\omega}}(x) \right] = d_2(p_{\boldsymbol{\omega}'} \| p_{\boldsymbol{\omega}}) - 1 \simeq \frac{\alpha}{2} (\boldsymbol{\omega}' - \boldsymbol{\omega})^T \mathcal{F}(\boldsymbol{\omega}) (\boldsymbol{\omega}' - \boldsymbol{\omega})$. It also justifies the use of natural gradients in off-distribution optimization, since a step in natural gradient direction has a controllable effect on the variance of the importance weights.

## 5 Policy Optimization via Importance Sampling

In this section, we discuss how to customize the bound provided in Theorem 4.1 for policy optimization, developing a novel model-free actor-only policy search algorithm, named *Policy Optimization via Importance Sampling* (POIS). We propose two versions of POIS: *Action-based POIS* (A-POIS), which is based on a policy gradient approach, and *Parameter-based POIS* (P-POIS), which adopts the PGPE framework. A more detailed description of the implementation aspects is reported in Appendix E.

### 5.1 Action-based POIS

In Action-based POIS (A-POIS) we search for a policy that maximizes the performance index $J_D(\boldsymbol{\theta})$ within a parametric space $\Pi_\Theta = \{\pi_{\boldsymbol{\theta}} : \boldsymbol{\theta} \in \Theta \subseteq \mathbb{R}^p\}$ of stochastic differentiable policies. In this context, the behavioral (resp. target) distribution $Q$ (resp. $P$) becomes the distribution over trajectories $p(\cdot|\boldsymbol{\theta})$ (resp. $p(\cdot|\boldsymbol{\theta}')$) induced by the behavioral policy $\pi_{\boldsymbol{\theta}}$ (resp. target policy $\pi_{\boldsymbol{\theta}'}$) and $f$ is the trajectory return $R(\tau)$ which is uniformly bounded as $|R(\tau)| \leq R_{\max} \frac{1 - \gamma^H}{1 - \gamma}$.[5] The surrogate loss function cannot be directly optimized via gradient ascent since computing $d_\alpha \left( p(\cdot|\boldsymbol{\theta}') \| p(\cdot|\boldsymbol{\theta}) \right)$ requires the approximation of an integral over the trajectory space and, for stochastic environments, to know the transition model $P$, which is unknown in a model-free setting. Simple bounds to this quantity, like $d_\alpha \left( p(\cdot|\boldsymbol{\theta}') \| p(\cdot|\boldsymbol{\theta}) \right) \leq \sup_{s \in \mathcal{S}} d_\alpha \left( \pi_{\boldsymbol{\theta}'}(\cdot|s) \| \pi_{\boldsymbol{\theta}}(\cdot|s) \right)^H$, besides being hard to compute due to the presence of the supremum, are extremely conservative since the Rényi divergence is raised to the horizon $H$. We suggest the replacement of the Rényi divergence with an estimate $\widehat{d}_2 \left( p(\cdot|\boldsymbol{\theta}') \| p(\cdot|\boldsymbol{\theta}) \right) = \frac{1}{N} \sum_{i=1}^N \prod_{t=0}^{H-1} d_2 \left( \pi_{\boldsymbol{\theta}'}(\cdot|s_{\tau_i,t}) \| \pi_{\boldsymbol{\theta}}(\cdot|s_{\tau_i,t}) \right)$ defined only in terms of the policy Rényi divergence (see Appendix E.2 for details). Thus, we obtain the following surrogate

objective:

$$\mathcal{L}_{\lambda}^{\text{A-POIS}}(\boldsymbol{\theta}'/\boldsymbol{\theta}) = \frac{1}{N}\sum_{i=1}^{N} w_{\boldsymbol{\theta}'/\boldsymbol{\theta}}(\tau_i)R(\tau_i) - \lambda\sqrt{\frac{\widehat{d_2}\left(p(\cdot|\boldsymbol{\theta}')\|p(\cdot|\boldsymbol{\theta})\right)}{N}}, \tag{9}$$

where $w_{\boldsymbol{\theta}'/\boldsymbol{\theta}}(\tau_i) = \frac{p(\tau_i|\boldsymbol{\theta}')}{p(\tau_i|\boldsymbol{\theta})} = \prod_{t=0}^{H-1}\frac{\pi_{\boldsymbol{\theta}'}(a_{\tau_i,t}|s_{\tau_i,t})}{\pi_{\boldsymbol{\theta}}(a_{\tau_i,t}|s_{\tau_i,t})}$. We consider the case in which $\pi_{\boldsymbol{\theta}}(\cdot|s)$ is a Gaussian distribution over actions whose mean depends on the state and whose covariance is state-independent and diagonal: $\mathcal{N}(u_{\boldsymbol{\mu}}(s), \text{diag}(\boldsymbol{\sigma}^2))$, where $\boldsymbol{\theta} = (\boldsymbol{\mu}, \boldsymbol{\sigma})$. The learning process mixes online and offline optimization. At each online iteration $j$, a dataset of $N$ trajectories is collected by executing in the environment the current policy $\pi_{\boldsymbol{\theta}_0^j}$. These trajectories are used to optimize the surrogate loss function $\mathcal{L}_{\lambda}^{\text{A-POIS}}$. At each offline iteration $k$, the parameters are updated via gradient ascent: $\boldsymbol{\theta}_{k+1}^j = \boldsymbol{\theta}_k^j + \alpha_k\mathcal{G}(\boldsymbol{\theta}_k^j)^{-1}\nabla_{\boldsymbol{\theta}_k^j}\mathcal{L}(\boldsymbol{\theta}_k^j/\boldsymbol{\theta}_0^j)$, where $\alpha_k > 0$ is the step size which is chosen via line search (see Appendix E.1) and $\mathcal{G}(\boldsymbol{\theta}_k^j)$ is a positive semi-definite matrix (e.g., $\mathcal{F}(\boldsymbol{\theta}_k^j)$, the FIM, for natural gradient)[6]. The pseudo-code of POIS is reported in Algorithm 1.

## 5.2  Parameter-based POIS

In the Parameter-based POIS (P-POIS) we again consider a parametrized policy space $\Pi_{\Theta} = \{\pi_{\boldsymbol{\theta}} : \boldsymbol{\theta} \in \Theta \subseteq \mathbb{R}^p\}$, but $\pi_{\boldsymbol{\theta}}$ needs not be differentiable. The policy parameters $\boldsymbol{\theta}$ are sampled at the beginning of each episode from a parametric hyperpolicy $\nu_{\boldsymbol{\rho}}$ selected in a parametric space $\mathcal{N}_{\mathcal{P}} = \{\nu_{\boldsymbol{\rho}} : \boldsymbol{\rho} \in \mathcal{P} \subseteq \mathbb{R}^r\}$. The goal is to learn the *hyperparameters* $\boldsymbol{\rho}$ so as to maximize $J_D(\boldsymbol{\rho})$. In this setting, the distributions $Q$ and $P$ of Section 4 correspond to the behavioral $\nu_{\boldsymbol{\rho}}$ and target $\nu_{\boldsymbol{\rho}'}$ hyperpolicies, while $f$ remains the trajectory return $R(\tau)$. The importance weights [67] must take into account all sources of randomness, derived from sampling a policy parameter $\boldsymbol{\theta}$ and a trajectory $\tau$: $w_{\boldsymbol{\rho}'/\boldsymbol{\rho}}(\boldsymbol{\theta}) = \frac{\nu_{\boldsymbol{\rho}'}(\boldsymbol{\theta})p(\tau|\boldsymbol{\theta})}{\nu_{\boldsymbol{\rho}}(\boldsymbol{\theta})p(\tau|\boldsymbol{\theta})} = \frac{\nu_{\boldsymbol{\rho}'}(\boldsymbol{\theta})}{\nu_{\boldsymbol{\rho}}(\boldsymbol{\theta})}$. In practice, a Gaussian hyperpolicy $\nu_{\boldsymbol{\rho}}$ with diagonal covariance matrix is often used, i.e., $\mathcal{N}(\boldsymbol{\mu}, \text{diag}(\boldsymbol{\sigma}^2))$ with $\boldsymbol{\rho} = (\boldsymbol{\mu}, \boldsymbol{\sigma})$. The policy is assumed to be deterministic: $\pi_{\boldsymbol{\theta}}(a|s) = \delta(a - u_{\boldsymbol{\theta}}(s))$, where $u_{\boldsymbol{\theta}}$ is a deterministic function of the state $s$ [e.g., 46, 14]. A first advantage over the action-based setting is that the distribution of the importance weights is entirely known, as it is the ratio of two Gaussians and the Rényi divergence $d_2(\nu_{\boldsymbol{\rho}'}\|\nu_{\boldsymbol{\rho}})$ can be computed exactly [6] (see Appendix C). This leads to the following surrogate objective:

$$\mathcal{L}_{\lambda}^{\text{P-POIS}}(\boldsymbol{\rho}'/\boldsymbol{\rho}) = \frac{1}{N}\sum_{i=1}^{N} w_{\boldsymbol{\rho}'/\boldsymbol{\rho}}(\boldsymbol{\theta}_i)R(\tau_i) - \lambda\sqrt{\frac{d_2\left(\nu_{\boldsymbol{\rho}'}\|\nu_{\boldsymbol{\rho}}\right)}{N}}, \tag{10}$$

where each trajectory $\tau_i$ is obtained by running an episode with action policy $\pi_{\boldsymbol{\theta}_i}$, and the corresponding policy parameters $\boldsymbol{\theta}_i$ are sampled independently from hyperpolicy $\nu_{\boldsymbol{\rho}}$, at the beginning of each episode. The hyperpolicy parameters are then updated offline as $\boldsymbol{\rho}_{k+1}^j = \boldsymbol{\rho}_k^j + \alpha_k\mathcal{G}(\boldsymbol{\rho}_k^j)^{-1}\nabla_{\boldsymbol{\rho}_k^j}\mathcal{L}(\boldsymbol{\rho}_k^j/\boldsymbol{\rho}_0^j)$ (see Algorithm 2 for the complete pseudo-code). A further advantage w.r.t. the action-based case is that the FIM $\mathcal{F}(\boldsymbol{\rho})$ can be computed exactly, and it is diagonal in the case of a Gaussian hyperpolicy with diagonal covariance matrix, turning a problematic inversion into a trivial division (the FIM is block-diagonal in the more general case of a Gaussian hyperpolicy, as observed in [26]). This makes natural gradient much more enticing for P-POIS.

# 6  Experimental Evaluation

In this section, we present the experimental evaluation of POIS in its two flavors (action-based and parameter-based). We first provide a set of empirical comparisons on classical continuous control tasks with linearly parametrized policies; we then show how POIS can be also adopted for learning deep neural policies. In all experiments, for the A-POIS we used the IS estimator, while for P-POIS we employed the SN estimator. All experimental details are provided in Appendix F.

## 6.1  Linear Policies

Linear parametrized Gaussian policies proved their ability to scale on complex control tasks [38]. In this section, we compare the learning performance of A-POIS and P-POIS against TRPO [42] and

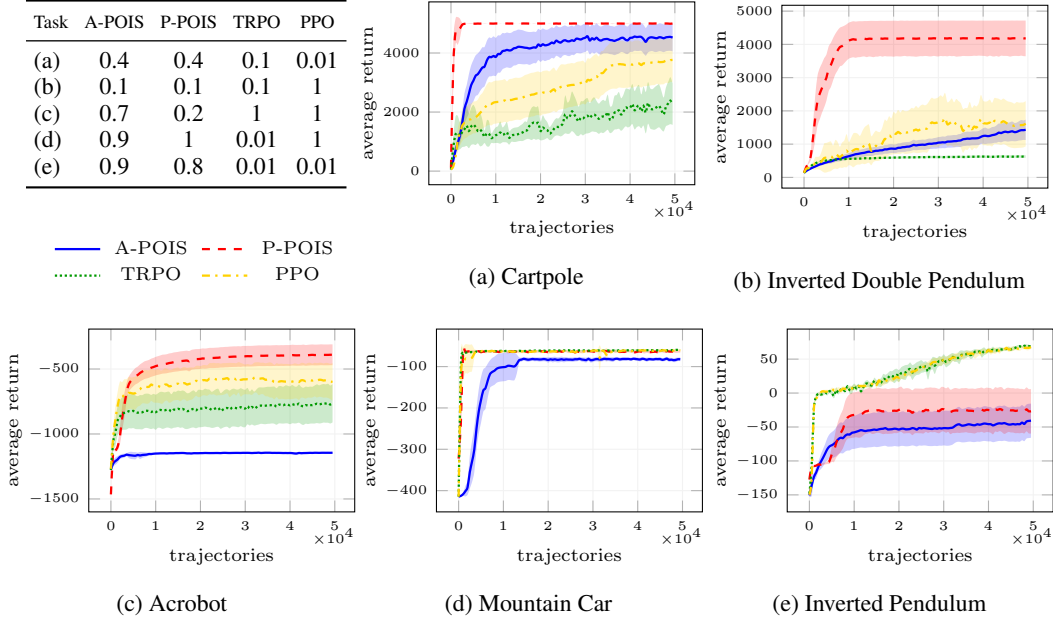

| Task | A-POIS | P-POIS | TRPO | PPO |
|------|--------|--------|------|-----|
| (a) | 0.4 | 0.4 | 0.1 | 0.01 |
| (b) | 0.1 | 0.1 | 0.1 | 1 |
| (c) | 0.7 | 0.2 | 1 | 1 |
| (d) | 0.9 | 1 | 0.01 | 1 |
| (e) | 0.9 | 0.8 | 0.01 | 0.01 |

(a) Cartpole

(b) Inverted Double Pendulum

(c) Acrobot

(d) Mountain Car

(e) Inverted Pendulum

Figure 1: Average return as a function of the number of trajectories for A-POIS, P-POIS and TRPO with *linear policy* (20 runs, 95% c.i.). The table reports the best hyperparameters found ($\delta$ for POIS and the step size for TRPO and PPO).

PPO [44] on classical continuous control benchmarks [12]. In Figure 1, we can see that both versions of POIS are able to significantly outperform both TRPO and PPO in the Cartpole environments, especially the P-POIS. In the Inverted Double Pendulum environment the learning curve of P-POIS is remarkable while A-POIS displays a behavior comparable to PPO. In the Acrobot task, P-POIS displays a better performance w.r.t. TRPO and PPO, but A-POIS does not keep up. In Mountain Car, we see yet another behavior: the learning curves of TRPO, PPO and P-POIS are almost one-shot (even if PPO shows a small instability), while A-POIS fails to display such a fast convergence. Finally, in the Inverted Pendulum environment, TRPO and PPO outperform both versions of POIS. This example highlights a limitation of our approach. Since POIS performs an importance sampling procedure at trajectory level, it cannot assign credit to good actions in bad trajectories. On the contrary, weighting each sample, TRPO and PPO are able also to exploit good trajectory segments. In principle, this problem can be mitigated in POIS by resorting to *per-decision importance sampling* [36], in which the weight is assigned to individual rewards instead of trajectory returns. Overall, POIS displays a performance comparable with TRPO and PPO across the tasks. In particular, P-POIS displays a better performance w.r.t. A-POIS. However, this ordering is not maintained when moving to more complex policy architectures, as shown in the next section.

In Figure 2 we show, for several metrics, the behavior of A-POIS when changing the $\delta$ parameter in the Cartpole environment. We can see that when $\delta$ is small (e.g., 0.2), the Effective Sample Size (ESS) remains large and, consequently, the variance of the importance weights ($\mathbb{Var}[w]$) is small. This means that the penalization term in the objective function discourages the optimization process from selecting policies which are far from the behavioral policy. As a consequence, the displayed behavior is very conservative, preventing the policy from reaching the optimum. On the contrary, when $\delta$ approaches 1, the ESS is smaller and the variance of the weights tends to increase significantly. Again, the performance remains suboptimal as the penalization term in the objective function is too light. The best behavior is obtained with an intermediate value of $\delta$, specifically 0.4.

## 6.2 Deep Neural Policies

In this section, we adopt a deep neural network (3 layers: 100, 50, 25 neurons each) to represent the policy. The experiment setup is fully compatible with the classical benchmark [12]. While A-POIS can be directly applied to deep neural networks, P-POIS exhibits some critical issues. A highly dimensional hyperpolicy (like a Gaussian from which the weights of an MLP policy are

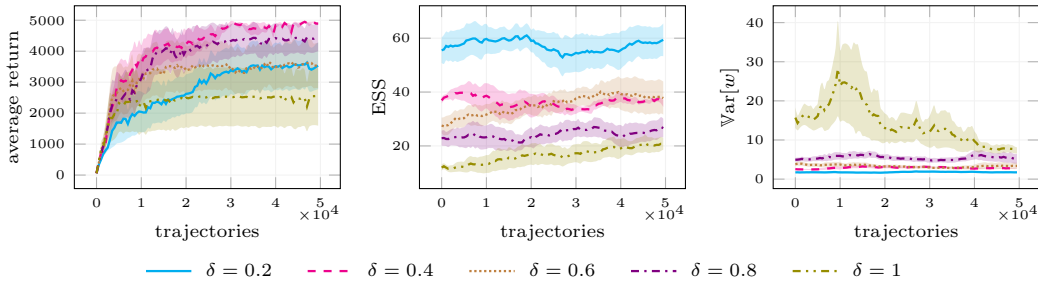

Figure 2: Average return, Effective Sample Size (ESS) and variance of the importance weights ($\mathbb{V}\text{ar}[w]$) as a function of the number of trajectories for A-POIS for different values of the parameter $\delta$ in the Cartpole environment (20 runs, 95% c.i.).

Table 1: Performance of POIS compated with [12] on *deep neural policies* (5 runs, 95% c.i.). In **bold**, the performances that are not statistically significantly different from the best algorithm in each task.

| Algorithm | Cart-Pole Balancing | Mountain Car | Double Inverted Pendulum | Swimmer |
|---|---|---|---|---|
| REINFORCE | $4693.7 \pm 14.0$ | $-67.1 \pm 1.0$ | $4116.5 \pm 65.2$ | $92.3 \pm 0.1$ |
| TRPO | $\mathbf{4869.8 \pm 37.6}$ | $\mathbf{-61.7 \pm 0.9}$ | $\mathbf{4412.4 \pm 50.4}$ | $\mathbf{96.0 \pm 0.2}$ |
| DDPG | $4634.4 \pm 87.6$ | $-288.4 \pm 170.3$ | $2863.4 \pm 154.0$ | $85.8 \pm 1.8$ |
| A-POIS | $\mathbf{4842.8 \pm 13.0}$ | $-63.7 \pm 0.5$ | $\mathbf{4232.1 \pm 189.5}$ | $88.7 \pm 0.55$ |
| CEM | $4815.4 \pm 4.8$ | $-66.0 \pm 2.4$ | $2566.2 \pm 178.9$ | $68.8 \pm 2.4$ |
| P-POIS | $4428.1 \pm 138.6$ | $-78.9 \pm 2.5$ | $3161.4 \pm 959.2$ | $76.8 \pm 1.6$ |

sampled) can make $d_2(\nu_{\boldsymbol{\rho}'} \| \nu_{\boldsymbol{\rho}})$ extremely sensitive to small parameter changes, leading to over-conservative updates.[7] A first practical variant comes from the insight that $d_2(\nu_{\boldsymbol{\rho}'} \| \nu_{\boldsymbol{\rho}})/N$ is the inverse of the effective sample size, as reported in Equation 6. We can obtain a less conservative (although approximate) surrogate function by replacing it with $1/\widehat{\text{ESS}}(\nu_{\boldsymbol{\rho}'} \| \nu_{\boldsymbol{\rho}})$. Another trick is to model the hyperpolicy as a set of independent Gaussians, each defined over a disjoint subspace of $\Theta$ (implementation details are provided in Appendix E.5). In Table 1, we augmented the results provided in [12] with the performance of POIS for the considered tasks. We can see that A-POIS is able to reach an overall behavior comparable with the best of the action-based algorithms, approaching TRPO and beating DDPG. Similarly, P-POIS exhibits a performance similar to CEM [52], the best performing among the parameter-based methods. The complete results are reported in Appendix F.

## 7 Discussion and Conclusions

In this paper, we presented a new actor-only policy optimization algorithm, POIS, which alternates online and offline optimization in order to efficiently exploit the collected trajectories, and can be used in combination with action-based and parameter-based exploration. In contrast to the state-of-the-art algorithms, POIS has a strong theoretical grounding, since its surrogate objective function derives from a statistical bound on the estimated performance, that is able to capture the uncertainty induced by importance sampling. The experimental evaluation showed that POIS, in both its versions (action-based and parameter-based), is able to achieve a performance comparable with TRPO, PPO and other classical algorithms on continuous control tasks. Natural extensions of POIS could focus on employing per-decision importance sampling, adaptive batch size, and trajectory reuse. Future work also includes scaling POIS to high-dimensional tasks and highly-stochastic environments. We believe that this work represents a valuable starting point for a deeper understanding of modern policy optimization and for the development of effective and scalable policy search methods.

## Acknowledgments

The study was partially funded by Lombardy Region (Announcement PORFESR 2014-2020). F. F. was partially funded through ERC Advanced Grant (no: 742870). We gratefully acknowledge the support of NVIDIA Corporation with the donation of the Tesla K40cm, Titan XP and Tesla V100 used for this research.

## Footnotes

[1]Other solutions to these problems have been proposed in the action-based literature, like the aforementioned DPG algorithm, the gradient baselines [34] and the actor-critic architectures [21].

[2]Note that this regularization term appears in the performance improvement bound, which contains exact quantities only. Thus, it does not really account for the uncertainty derived from the importance sampling.

[3]Note that $\left| \widetilde{\mu}_{P/Q} \right| \le \|f\|_\infty$. Therefore, its variance is always finite.

[4]Although the variance tends to be reduced in the learning process, there might be cases in which it needs to be increased (e.g., suppose we start with a behavioral policy with small variance, it might be beneficial increasing the variance to enforce exploration).

[5] When $\gamma \to 1$ the bound becomes $H R_{\max}$.

[6]The FIM needs to be estimated via importance sampling as well, as shown in Appendix E.3.

[7]This curse of dimensionality, related to $\dim(\boldsymbol{\theta})$, has some similarities with the dependence of the Rényi divergence on the actual horizon $H$ in the action-based case.

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
