[Supplementary Material]

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

## Index of the Appendix

In the following, we briefly recap the contents of the Appendix.

## A  Related Works

Policy optimization algorithms can be classified according to different dimensions (Table 2). It is by now established, in the policy-based RL community, that effective algorithms, either on-policy or off-policy, should account for the variance of the gradient estimate. Early attempts, in the class of action-based algorithms, are the usage of a baseline to reduce the estimated gradient variance without introducing bias [4, 34]. A similar rationale is at the basis of actor-critic architectures [21, 51, 33], in which an estimate of the value function is used to reduce uncertainty. Baselines are typically constant (REINFORCE), time-dependent (G(PO)MDP) or state-dependent (actor-critic), but these approaches have been recently extended to account for action-dependent baselines [58, 65]. Even though parameter-based algorithms are, by nature, affected by smaller variance w.r.t. action-based ones, similar baselines can be derived [66]. A first dichotomy in the class of policy-based algorithms comes when considering the minimal unit used to compute the gradient. *Episode-based* (or episodic) approaches [e.g., 64, 4] perform the gradient estimation by averaging the gradients of each episode which need to have a finite horizon. On the contrary, *step-based* approaches [e.g., 42, 44, 23], derived from the Policy Gradient Theorem [51], can estimate the gradient by averaging over timesteps. The latter requires a function approximator (a critic) to estimate the Q-function, or directly the advantage function [43]. When coming to the on/off-policy dichotomy, the previous distinction has a relevant impact. Indeed, episode-based approaches need to perform importance sampling on trajectories, thus the importance weights are the products of policy ratios for all executed actions within a trajectory, whereas step-based algorithms need just to weight each sample with the corresponding policy ratio. The latter case helps to keep the value of the importance weights close to one, but the need to have a critic prevents from a complete analysis of the uncertainty since the bias/variance injected by the critic is hard to compute [21]. Moreover, in the off-policy scenario, it is necessary to control some notion of dissimilarity between the behavioral and target policy, as the variance increases when moving too far. This is the case of TRPO [42], where the regularization constraint based on the Kullback-Leibler divergence helps controlling the importance weights but originates from an exact bound on the performance improvement. Intuitively, the same rationale applies to the truncation of the importance weights, employed by PPO, that avoids performing too large steps in the policy space. Nevertheless, the step size in TRPO and the truncation range $\epsilon$ in PPO are just hyperparameters and have a limited statistical meaning. On the contrary, other actor-critic architectures have been proposed including also experience replay methods, like [61] in which the importance weights are truncated, but the method is able to account for the injected bias. The authors propose to keep a running mean of the best policies seen so far to avoid a hard constraint on the policy dissimilarity. Differently from these methods, POIS directly models the uncertainty due to the importance sampling procedure. The bound in Theorem 4.1 introduces the unique hyperparameter $\delta$ which has a precise statistical meaning as confidence level. The optimal value of $\delta$ (like the step size in TRPO and $\epsilon$ in PPO) is task-dependent and might vary during the learning procedure. Furthermore, POIS is an episode-based approach in which the importance weights account for the whole trajectory at once; this might prevent from assigning credit to valuable subtrajectories (like in the case of Inverted Pendulum, see Figure 1). A possible solution is to resort to per-decision importance sampling [36].

Table 2: Comparison of some policy optimization algorithms according to different dimensions. For brevity, we will indicate with $w_{\theta'/\theta}(a_t|s_t) = \frac{\pi_{\theta'}(a_t|s_t)}{\pi_{\theta}(a_t|s_t)}$. For episode-based algorithms we will indicate with $\widehat{\mathbb{E}}_{\tau\sim\theta}$ the empirical average over trajectories collected with $\pi_\theta$. For step-based algorithms $\widehat{\mathbb{E}}_{t\sim\theta}$ is the empirical average collecting samples with $\pi_\theta$. For parameter-based algorithms we indicate with $\widehat{\mathbb{E}}_{\theta\sim\rho,\tau\sim\theta}$ the empirical expectation taken w.r.t. policy parameter $\boldsymbol{\theta}$ sampled from the hyperpolicy $\nu_\rho$ and trajectory $\tau$ collected with $\pi_{\boldsymbol\theta}$. For the actor-critic architectures, $\widehat{Q}$ and $\widehat{A}$ are the estimated Q-function and advantage function.

| Algorithm | Action/Parameter based | On/Off policy | Optimization problem | Critic | Timestep/Trajectory based |
|---|---|---|---|---|---|
| REINFORCE/ G(PO)MDP [64, 4] | action-based | on-policy | $\max \widehat{\mathbb{E}}_{\tau\sim\boldsymbol\theta}[R(\tau)]$ | No | episode-based |
| TRPO [42] | action-based | on-policy | $\max \widehat{\mathbb{E}}_{t\sim\boldsymbol\theta}\left[w_{\theta'/\theta}(a_t|s_t)\widehat{A}(s_t,a_t)\right]$ s.t. $\widehat{\mathbb{E}}_{t\sim\boldsymbol\theta}[D_{\mathrm{KL}}(\pi_{\boldsymbol\theta'}(\cdot|s_t)\|\pi_{\boldsymbol\theta}(\cdot|s_t))] \leq \delta$ | Yes | step-based |
| PPO [44] | action-based | on/off-policy | $\max \widehat{\mathbb{E}}_{t\sim\boldsymbol\theta}\left[\min\left\{w_{\theta'/\theta}(a_t|s_t)\widehat{A}(s_t,a_t),\right.\right.$ $\left.\left. \mathrm{clip}\left(w_{\theta'/\theta}(a_t|s_t),1-\epsilon,1+\epsilon\right)\widehat{A}(s_t,a_t)\right\}\right]$ | Yes | step-based |
| DDPG [23] | action-based | off-policy | $\max \widehat{\mathbb{E}}_{t\sim\boldsymbol\theta}\left[\pi_{\boldsymbol\theta'}(a_t|s_t)\widehat{Q}(s_t,a_t)\right]$ | Yes | step-based |
| REPS [31][8] | action-based | on-policy | $\max \widehat{\mathbb{E}}_{t\sim\boldsymbol\theta}[R(s_t,a_t)]$ s.t. $\widehat{\mathbb{E}}_{t\sim\boldsymbol\theta}\left[D_{\mathrm{KL}}(d_\mu^{\pi_{\theta'}}(s_t,a_t)\|d_\mu^{\pi_{\boldsymbol\theta}}(s_t,a_t))\right] \leq \delta$ | Yes | step-based |
| RWR [32] | action-based | on-policy | $\max \widehat{\mathbb{E}}_{t\sim\boldsymbol\theta}[\beta\exp(-\beta R(s_t,a_t))]$ | No | step-based |
| A-POIS | action-based | on/off-policy | $\max \widehat{\mathbb{E}}_{\tau\sim\boldsymbol\theta}\left[w_{\theta'/\theta}(\tau)R(\tau)\right] - \lambda\sqrt{\widehat{d_2(p(\cdot|\boldsymbol\theta')\|p(\cdot|\boldsymbol\theta))}/N}$ | No | episode-based |
| PGPE [45] | parameter-based | on-policy | $\max \widehat{\mathbb{E}}_{\boldsymbol\theta\sim\rho,\tau\sim\boldsymbol\theta}[R(\tau)]$ | No | episode-based |
| IW-PGPE [67] | parameter-based | on/off-policy | $\max \widehat{\mathbb{E}}_{\boldsymbol\theta\sim\rho,\tau\sim\boldsymbol\theta}[w_{\rho'/\rho}(\boldsymbol\theta)R(\tau)]$ | No | episode-based |
| P-POIS | parameter-based | on/off-policy | $\max \widehat{\mathbb{E}}_{\boldsymbol\theta\sim\rho,\tau\sim\boldsymbol\theta}\left[w_{\rho'/\rho}(\boldsymbol\theta)R(\tau)\right] - \lambda\sqrt{\widehat{d_2(\nu_\rho\|\nu_\rho)}/N}$ | No | episode-based |

## B   Proofs and Derivations

**Lemma 4.1.** *Let $P$ and $Q$ be two probability measures on the measurable space $(\mathcal{X}, \mathcal{F})$ such that $P \ll Q$. Let $\mathbf{x} = (x_1, x_2, \ldots, x_N)^T$ i.i.d. random variables sampled from $Q$ and $f : \mathcal{X} \to \mathbb{R}$ be a bounded function ($\|f\|_\infty < +\infty$). Then, for any $N > 0$, the variance of the IS estimator $\widehat{\mu}_{P/Q}$ can be upper bounded as:*

$$\operatorname*{Var}_{\mathbf{x} \sim Q} \left[ \widehat{\mu}_{P/Q} \right] \leq \frac{1}{N} \|f\|_\infty^2 d_2 \left( P \| Q \right). \tag{7}$$

*Proof.* From the fact that $x_i$ are i.i.d. we can write:

$$\operatorname*{Var}_{\mathbf{x} \sim Q} \left[ \widehat{\mu}_{P/Q} \right] \leq \frac{1}{N} \operatorname*{Var}_{x_1 \sim Q} \left[ \frac{p(x_1)}{q(x_1)} f(x_1) \right] \leq \frac{1}{N} \operatorname*{\mathbb{E}}_{x_1 \sim Q} \left[ \left( \frac{p(x_1)}{q(x_1)} f(x_1) \right)^2 \right]$$

$$\leq \frac{1}{N} \|f\|_\infty^2 \operatorname*{\mathbb{E}}_{x_1 \sim Q} \left[ \left( \frac{p(x_1)}{q(x_1)} \right)^2 \right] = \frac{1}{N} \|f\|_\infty^2 d_2 \left( P \| Q \right).$$

$\square$

**Theorem 4.1.** *Let $P$ and $Q$ be two probability measures on the measurable space $(\mathcal{X}, \mathcal{F})$ such that $P \ll Q$ and $d_2(P\|Q) < +\infty$. Let $x_1, x_2, \ldots, x_N$ be i.i.d. random variables sampled from $Q$, and $f : \mathcal{X} \to \mathbb{R}$ be a bounded function ($\|f\|_\infty < +\infty$). Then, for any $0 < \delta \leq 1$ and $N > 0$ with probability at least $1 - \delta$ it holds that:*

$$\operatorname*{\mathbb{E}}_{x \sim P} [f(x)] \geq \frac{1}{N} \sum_{i=1}^{N} w_{P/Q}(x_i) f(x_i) - \|f\|_\infty \sqrt{\frac{(1-\delta) d_2(P\|Q)}{\delta N}}. \tag{8}$$

*Proof.* We start from Cantelli's inequality applied on the random variable $\widehat{\mu}_{P/Q} = \frac{1}{N} \sum_{i=1}^{N} w_{P/Q}(x_i) f(x_i)$:

$$\Pr \left( \widehat{\mu}_{P/Q} - \operatorname*{\mathbb{E}}_{x \sim P} [f(x)] \geq \lambda \right) \leq \frac{1}{1 + \frac{\lambda^2}{\operatorname{Var}_{\mathbf{x} \sim Q} [\widehat{\mu}_{P/Q}]}}. \tag{11}$$

By calling $\delta = \frac{1}{1 + \frac{\lambda^2}{\operatorname{Var}_{\mathbf{x} \sim Q} [\widehat{\mu}_{P/Q}]}}$ and considering the complementary event, we get that with probability at least $1 - \delta$ we have:

$$\operatorname*{\mathbb{E}}_{x \sim P} [f(x)] \geq \widehat{\mu}_{P/Q} - \sqrt{\frac{1-\delta}{\delta} \operatorname*{Var}_{\mathbf{x} \sim Q} \left[ \widehat{\mu}_{P/Q} \right]}. \tag{12}$$

By replacing the variance with the bound in Theorem 4.1 we get the result. $\square$

**Theorem 4.2.** *Let $p_{\boldsymbol{\omega}}$ be a p.d.f. differentiable w.r.t. $\boldsymbol{\omega} \in \Omega$. Then, it holds that, for the Rényi divergence: $D_\alpha(p_{\boldsymbol{\omega}'} \| p_{\boldsymbol{\omega}}) = \frac{\alpha}{2} (\boldsymbol{\omega}' - \boldsymbol{\omega})^T \mathcal{F}(\boldsymbol{\omega}) (\boldsymbol{\omega}' - \boldsymbol{\omega}) + o(\|\boldsymbol{\omega}' - \boldsymbol{\omega}\|_2^2)$, and for the exponentiated Rényi divergence: $d_\alpha(p_{\boldsymbol{\omega}'} \| p_{\boldsymbol{\omega}}) = 1 + \frac{\alpha}{2} (\boldsymbol{\omega}' - \boldsymbol{\omega})^T \mathcal{F}(\boldsymbol{\omega}) (\boldsymbol{\omega}' - \boldsymbol{\omega}) + o(\|\boldsymbol{\omega}' - \boldsymbol{\omega}\|_2^2)$.*

*Proof.* We need to compute the second-order Taylor expansion of the $\alpha$-Rényi divergence. We start considering the term:

$$I(\boldsymbol{\omega}') = \int_{\mathcal{X}} \left( \frac{p_{\boldsymbol{\omega}'}(x)}{p_{\boldsymbol{\omega}}(x)} \right)^\alpha p_{\boldsymbol{\omega}}(x) \, \mathrm{d}x = \int_{\mathcal{X}} p_{\boldsymbol{\omega}'}(x)^\alpha p_{\boldsymbol{\omega}}(x)^{1-\alpha} \, \mathrm{d}x. \tag{13}$$

The gradient is given by:

$$\nabla_{\boldsymbol{\omega}'} I(\boldsymbol{\omega}') = \int_{\mathcal{X}} \nabla_{\boldsymbol{\omega}'} p_{\boldsymbol{\omega}'}(x)^\alpha p_{\boldsymbol{\omega}}(x)^{1-\alpha} \, \mathrm{d}x = \alpha \int_{\mathcal{X}} p_{\boldsymbol{\omega}'}(x)^{\alpha-1} p_{\boldsymbol{\omega}}(x)^{1-\alpha} \nabla_{\boldsymbol{\omega}'} p_{\boldsymbol{\omega}'}(x) \, \mathrm{d}x.$$

Thus, $\nabla_{\boldsymbol{\omega}'} I(\boldsymbol{\omega}')|_{\boldsymbol{\omega}'=\boldsymbol{\omega}} = \mathbf{0}$. We now compute the Hessian:

$$\mathcal{H}_{\boldsymbol{\omega}'} I(\boldsymbol{\omega}') = \nabla_{\boldsymbol{\omega}'} \nabla_{\boldsymbol{\omega}'}^T I(\boldsymbol{\omega}') = \alpha \nabla_{\boldsymbol{\omega}'} \int_{\mathcal{X}} p_{\boldsymbol{\omega}'}(x)^{\alpha-1} p_{\boldsymbol{\omega}}(x)^{1-\alpha} \nabla_{\boldsymbol{\omega}'}^T p_{\boldsymbol{\omega}'}(x) \, \mathrm{d}x$$

$$= \alpha \int_{\mathcal{X}} \left( (\alpha-1) p_{\boldsymbol{\omega}'}(x)^{\alpha-2} p_{\boldsymbol{\omega}}(x)^{1-\alpha} \nabla_{\boldsymbol{\omega}'} p_{\boldsymbol{\omega}'}(x) \nabla_{\boldsymbol{\omega}'}^T p_{\boldsymbol{\omega}'}(x) + p_{\boldsymbol{\omega}'}(x)^{\alpha-1} p_{\boldsymbol{\omega}}(x)^{1-\alpha} \mathcal{H}_{\boldsymbol{\omega}'} p_{\boldsymbol{\omega}'}(x) \right) \, \mathrm{d}x.$$

Evaluating the Hessian in $\boldsymbol{\omega}$ we have:

$$\mathcal{H}_{\boldsymbol{\omega}'} I(\boldsymbol{\omega}')|_{\boldsymbol{\omega}'=\boldsymbol{\omega}} = \alpha(\alpha-1) \int_{\mathcal{X}} p_{\boldsymbol{\omega}}(x)^{-1} \nabla_{\boldsymbol{\omega}} p_{\boldsymbol{\omega}}(x) \nabla_{\boldsymbol{\omega}}^T p_{\boldsymbol{\omega}}(x) \, \mathrm{d}x$$

$$= \alpha(\alpha - 1) \int_{\mathcal{X}} p_{\boldsymbol{\omega}}(x) \nabla_{\boldsymbol{\omega}} \log p_{\boldsymbol{\omega}}(x) \nabla_{\boldsymbol{\omega}}^T \log p_{\boldsymbol{\omega}}(x) \, \mathrm{d}x = \alpha(\alpha - 1)\mathcal{F}(\boldsymbol{\omega}).$$

Now, $D_\alpha(p_{\boldsymbol{\omega}'} \| p_{\boldsymbol{\omega}}) = \frac{1}{\alpha - 1} \log I(\boldsymbol{\omega}')$. Thus:

$$\nabla_{\boldsymbol{\omega}'} D_\alpha(p_{\boldsymbol{\omega}'} \| p_{\boldsymbol{\omega}})|_{\boldsymbol{\omega}'=\boldsymbol{\omega}} = \frac{1}{\alpha - 1} \frac{\nabla_{\boldsymbol{\omega}'} I(\boldsymbol{\omega}')}{I(\boldsymbol{\omega}')}\bigg|_{\boldsymbol{\omega}'=\boldsymbol{\omega}} = \mathbf{0},$$

$$\mathcal{H}_{\boldsymbol{\omega}'} D_\alpha(p_{\boldsymbol{\omega}'} \| p_{\boldsymbol{\omega}})|_{\boldsymbol{\omega}'=\boldsymbol{\omega}} = \frac{1}{\alpha - 1} \frac{I(\boldsymbol{\omega}')\mathcal{H}_{\boldsymbol{\omega}'} I(\boldsymbol{\omega}') + \nabla_{\boldsymbol{\omega}'} I(\boldsymbol{\omega}') \nabla_{\boldsymbol{\omega}'}^T I(\boldsymbol{\omega}')}{(I(\boldsymbol{\omega}'))^2}\bigg|_{\boldsymbol{\omega}'=\boldsymbol{\omega}}$$

$$= \frac{1}{\alpha - 1} \mathcal{H}_{\boldsymbol{\omega}'} I(\boldsymbol{\omega}')|_{\boldsymbol{\omega}'=\boldsymbol{\omega}} = \alpha \mathcal{F}(\boldsymbol{\omega}),$$

having observed that $I(\boldsymbol{\omega}) = 1$. For what concerns the $d_\alpha(p_{\boldsymbol{\omega}'} \| p_{\boldsymbol{\omega}})$, we have:

$$\nabla_{\boldsymbol{\omega}'} d_\alpha(p_{\boldsymbol{\omega}'} \| p_{\boldsymbol{\omega}})|_{\boldsymbol{\omega}'=\boldsymbol{\omega}} = \nabla_{\boldsymbol{\omega}'} \exp\left(D_\alpha(p_{\boldsymbol{\omega}'} \| p_{\boldsymbol{\omega}})\right)|_{\boldsymbol{\omega}'=\boldsymbol{\omega}}$$

$$= \exp\left(D_\alpha(p_{\boldsymbol{\omega}'} \| p_{\boldsymbol{\omega}})\right) \nabla_{\boldsymbol{\omega}'} D_\alpha(p_{\boldsymbol{\omega}'} \| p_{\boldsymbol{\omega}})|_{\boldsymbol{\omega}'=\boldsymbol{\omega}} = \mathbf{0},$$

$$\mathcal{H}_{\boldsymbol{\omega}'} d_\alpha(p_{\boldsymbol{\omega}'} \| p_{\boldsymbol{\omega}})|_{\boldsymbol{\omega}'=\boldsymbol{\omega}} = \mathcal{H}_{\boldsymbol{\omega}'} \exp\left(D_\alpha(p_{\boldsymbol{\omega}'} \| p_{\boldsymbol{\omega}})\right)|_{\boldsymbol{\omega}'=\boldsymbol{\omega}}$$

$$= \exp\left(D_\alpha(p_{\boldsymbol{\omega}'} \| p_{\boldsymbol{\omega}})\right) \left(\mathcal{H}_{\boldsymbol{\omega}'} D_\alpha(p_{\boldsymbol{\omega}'} \| p_{\boldsymbol{\omega}}) + \nabla_{\boldsymbol{\omega}'} D_\alpha(p_{\boldsymbol{\omega}'} \| p_{\boldsymbol{\omega}}) \nabla_{\boldsymbol{\omega}'}^T D_\alpha(p_{\boldsymbol{\omega}'} \| p_{\boldsymbol{\omega}})\right)|_{\boldsymbol{\omega}'=\boldsymbol{\omega}}$$

$$= \alpha \mathcal{F}(\boldsymbol{\omega}).$$

$\square$

## C   Analysis of the IS estimator

In this Appendix, we analyze the behavior of the importance weights when the behavioral and target distributions are Gaussians. We start providing a closed-form expression for the Rényi divergence between multivariate Gaussian distributions [6]. Let $P \sim \mathcal{N}(\boldsymbol{\mu}_P, \boldsymbol{\Sigma}_P)$ and $Q \sim \mathcal{N}(\boldsymbol{\mu}_Q, \boldsymbol{\Sigma}_Q)$ and $\alpha \in [0, \infty]$:

$$D_\alpha(P \| Q) = \frac{\alpha}{2} (\boldsymbol{\mu}_P - \boldsymbol{\mu}_Q)^T \boldsymbol{\Sigma}_\alpha^{-1} (\boldsymbol{\mu}_P - \boldsymbol{\mu}_Q) - \frac{1}{2(\alpha - 1)} \log \frac{\det(\boldsymbol{\Sigma}_\alpha)}{\det(\boldsymbol{\Sigma}_P)^{1-\alpha} \det(\boldsymbol{\Sigma}_Q)^\alpha}, \quad (14)$$

where $\boldsymbol{\Sigma}_\alpha = \alpha \boldsymbol{\Sigma}_Q + (1 - \alpha)\boldsymbol{\Sigma}_P$ under the assumption that $\boldsymbol{\Sigma}_\alpha$ is positive-definite.

From now on, we will focus on univariate Gaussian distributions and we provide a closed-form expression for the importance weights and their probability density function $f_w$. We consider $Q \sim \mathcal{N}(\mu_Q, \sigma_Q^2)$ as behavioral distribution and $P \sim \mathcal{N}(\mu_P, \sigma_P^2)$ as target distribution. We assume that $\sigma_Q^2, \sigma_P^2 > 0$ and we consider the two cases: unequal variances and equal variances. For brevity, we will indicate with $w(x)$ the weight $w_{P/Q}(x)$.

### C.1   Unequal variances

When $\sigma_Q^2 \neq \sigma_P^2$, the expression of the importance weights is given by:

$$w(x) = \frac{\sigma_Q}{\sigma_P} \exp\left(\frac{1}{2} \frac{(\mu_P - \mu_Q)^2}{\sigma_Q^2 - \sigma_P^2}\right) \exp\left(-\frac{1}{2} \frac{\sigma_Q^2 - \sigma_P^2}{\sigma_Q^2 \sigma_P^2} \left(x - \frac{\sigma_Q^2 \mu_P - \sigma_P^2 \mu_Q}{\sigma_Q^2 - \sigma_P^2}\right)^2\right), \quad (15)$$

for $x \sim Q$. Let us first notice two distinct situations: if $\sigma_Q^2 - \sigma_P^2 > 0$ the weight $w(x)$ is upper bounded by $A = \frac{\sigma_Q}{\sigma_P} \exp\left(\frac{1}{2} \frac{(\mu_P - \mu_Q)^2}{\sigma_Q^2 - \sigma_P^2}\right)$, whereas if $\sigma_Q^2 - \sigma_P^2 < 0$, $w(x)$ is unbounded but it admits a minimum of value $A$. Let us investigate the probability density function.

**Proposition C.1.** *Let $Q \sim \mathcal{N}(\mu_Q, \sigma_Q^2)$ be the behavioral distribution and $P \sim \mathcal{N}(\mu_P, \sigma_P^2)$ be the target distribution, with $\sigma_Q^2 \neq \sigma_P^2$. The probability density function of $w(x) = p(x)/q(x)$ is given by:*

$$f_w(y) = \begin{cases} \frac{\overline{\sigma}}{y\sqrt{\pi \log \frac{A}{y}}} \exp\left(-\frac{1}{2}\overline{\mu}^2\right) \left(\frac{y}{A}\right)^{\overline{\sigma}^2} \cosh\left(\overline{\mu}\overline{\sigma}\sqrt{2\log\frac{A}{y}}\right), & \text{if } \sigma_Q^2 > \sigma_P^2, \ y \in [0, A], \\[2ex] \frac{\overline{\sigma}}{y\sqrt{\pi \log \frac{y}{A}}} \exp\left(-\frac{1}{2}\overline{\mu}^2\right) \left(\frac{A}{y}\right)^{\overline{\sigma}^2} \cosh\left(\overline{\mu}\overline{\sigma}\sqrt{2\log\frac{y}{A}}\right), & \text{if } \sigma_Q^2 < \sigma_P^2, \ y \in [A, \infty), \end{cases}$$

*where $\overline{\mu} = \frac{\sigma_Q}{\sigma_Q^2 - \sigma_P^2}(\mu_P - \mu_Q)$ and $\overline{\sigma}^2 = \frac{\sigma_P^2}{|\sigma_Q^2 - \sigma_P^2|}$.*

*Proof.* We look at $w(x)$ as a function of random variable $x \sim Q$. We introduce the following symbols:

$$m = \frac{\sigma_Q^2 \mu_P - \sigma_P^2 \mu_Q}{\sigma_Q^2 - \sigma_P^2}, \quad \tau = \frac{\sigma_Q^2 - \sigma_P^2}{\sigma_Q^2 \sigma_P^2}.$$

Let us start computing the c.d.f.:

$$F_w(y) = \Pr(w(x) \le y) = \Pr\left(A \exp\left(-\frac{1}{2}\tau(x-m)^2\right) \le y\right) = \Pr\left(\tau(x-m)^2 \ge -2\log\frac{y}{A}\right).$$

We distinguish the two cases according to the sign of $\tau$ and we observe that $x = \mu_Q + \sigma_Q z$ where $z \sim \mathcal{N}(0,1)$:

**$\tau > 0$:**

$$F_w(y) = \Pr\left((x-m)^2 \ge \frac{2}{\tau}\log\frac{A}{y}\right)$$

$$= \Pr\left(x \le m - \sqrt{\frac{2}{\tau}\log\frac{A}{y}}\right) + \Pr\left(x \ge m + \sqrt{\frac{2}{\tau}\log\frac{A}{y}}\right)$$

$$= \Pr\left(z \le \frac{m-\mu_Q}{\sigma_Q} - \sqrt{\frac{2}{\tau\sigma_Q^2}\log\frac{A}{y}}\right) + \Pr\left(z \ge \frac{m-\mu_Q}{\sigma_Q} + \sqrt{\frac{2}{\tau\sigma_Q^2}\log\frac{A}{y}}\right).$$

We call $\bar{\mu} = \frac{m-\mu_Q}{\sigma_Q} = \frac{\sigma_Q}{\sigma_Q^2 - \sigma_P^2}(\mu_P - \mu_Q)$ and $\bar{\sigma}^2 = \frac{1}{\tau\sigma_Q^2} = \frac{\sigma_P^2}{\sigma_Q^2 - \sigma_P^2}$, thus we have:

$$F_w(y) = \Pr\left(z \le \bar{\mu} - \sqrt{2\bar{\sigma}^2\log\frac{A}{y}}\right) + \Pr\left(z \ge \bar{\mu} + \sqrt{2\bar{\sigma}^2\log\frac{A}{y}}\right)$$

$$= \Phi\left(\bar{\mu} - \sqrt{2\bar{\sigma}^2\log\frac{A}{y}}\right) + 1 - \Phi\left(\bar{\mu} + \sqrt{2\bar{\sigma}^2\log\frac{A}{y}}\right),$$

where $\Phi$ is the c.d.f. of a normal standard distribution. By taking the derivative w.r.t. $y$ we get the p.d.f.:

$$f_w(y) = \frac{\partial F_w(y)}{\partial y} = -\sqrt{2\bar{\sigma}^2}\frac{1}{2\sqrt{\log\frac{A}{y}}}\frac{y}{A}\frac{-A}{y^2}\left(\phi\left(\bar{\mu} - \sqrt{2\bar{\sigma}^2\log\frac{A}{y}}\right) + \phi\left(\bar{\mu} + \sqrt{2\bar{\sigma}^2\log\frac{A}{y}}\right)\right)$$

$$= \frac{\sqrt{2\bar{\sigma}}}{2y\sqrt{\log\frac{A}{y}}}\left(\phi\left(\bar{\mu} - \sqrt{2\bar{\sigma}^2\log\frac{A}{y}}\right) + \phi\left(\bar{\mu} + \sqrt{2\bar{\sigma}^2\log\frac{A}{y}}\right)\right)$$

$$= \frac{\sqrt{2\bar{\sigma}}}{2y\sqrt{\log\frac{A}{y}}}\frac{1}{\sqrt{2\pi}}\left(\exp\left(-\frac{1}{2}\left(\bar{\mu} - \sqrt{2\bar{\sigma}^2\log\frac{A}{y}}\right)^2\right) + \exp\left(-\frac{1}{2}\left(\bar{\mu} + \sqrt{2\bar{\sigma}^2\log\frac{A}{y}}\right)^2\right)\right)$$

$$= \frac{\bar{\sigma}}{y\sqrt{\pi\log\frac{A}{y}}}\exp\left(-\frac{1}{2}\bar{\mu}^2\right)\exp\left(-\bar{\sigma}^2\log\frac{A}{y}\right)\frac{\exp\left(\bar{\mu}\bar{\sigma}\sqrt{2\log\frac{A}{y}}\right) + \exp\left(-\bar{\mu}\bar{\sigma}\sqrt{2\log\frac{A}{y}}\right)}{2}$$

$$= \frac{\bar{\sigma}}{y\sqrt{\pi\log\frac{A}{y}}}\exp\left(-\frac{1}{2}\bar{\mu}^2\right)\left(\frac{y}{A}\right)^{\bar{\sigma}^2}\cosh\left(\bar{\mu}\bar{\sigma}\sqrt{2\log\frac{A}{y}}\right),$$

where $\phi$ is the p.d.f. of a normal standard distribution.

**$\tau < 0$:** The derivation takes similar steps, all it takes is to call $\bar{\sigma}^2 = -\frac{1}{\tau\sigma_Q^2} = \frac{\sigma_P^2}{\sigma_P^2 - \sigma_Q^2}$, then the c.d.f. becomes:

$$F_w(y) = \Phi\left(\bar{\mu} + \sqrt{2\bar{\sigma}^2\log\frac{y}{A}}\right) - \Phi\left(\bar{\mu} - \sqrt{2\bar{\sigma}^2\log\frac{y}{A}}\right),$$

and the p.d.f. is:

$$f_w(x) = \frac{\bar{\sigma}}{y\sqrt{\pi\log\frac{y}{A}}}\exp\left(-\frac{1}{2}\bar{\mu}^2\right)\left(\frac{A}{y}\right)^{\bar{\sigma}^2}\cosh\left(\bar{\mu}\bar{\sigma}\sqrt{2\log\frac{y}{A}}\right).$$

To unify the two cases we set $\bar{\sigma}^2 = \frac{\sigma_P^2}{|\sigma_Q^2 - \sigma_P^2|}$. $\square$

It is interesting to investigate the properties of the tail of the distribution when $w$ is unbounded. Indeed, we discover that the distribution displays a fat-tail behavior.

**Proposition C.2.** *If $\sigma_P^2 > \sigma_Q^2$ then there exists $c > 0$ and $y_0 > 0$ such that for any $y \geq y_0$, the p.d.f. $f_w$ can be lower bounded as $f_w(y) \geq cy^{-1-\overline{\sigma}^2}(\log y)^{-\frac{1}{2}}$.*

*Proof.* Let us call $z = y/A$ and let $a > 0$ be a constant, then it holds that for sufficiently large $y$ we have:

$$f_w(y) \geq az^{-1-\overline{\sigma}^2}(\log z)^{-1/2}\exp\left(\sqrt{\log z}\right)^{\sqrt{2\mu\sigma}}. \tag{16}$$

To get the result, we observe that for $z > 1$ we have $\exp\left(\sqrt{\log z}\right) \geq 1$. Now, by replacing $z$ with $y/A$ we just need to change the constant $a$ into $c > 0$. $\square$

As a consequence, the $\alpha$-th moment of $w(x)$ does not exist for $\alpha - 1 - \overline{\sigma}^2 \geq -1 \implies \alpha \geq \overline{\sigma}^2 = \frac{\sigma_P^2}{\sigma_P^2 - \sigma_Q^2}$, this prevents from using Bernstein-like inequalities for bounding in probability the importance weights. The non-existence of finite moments is confirmed by the $\alpha$-Rényi divergence. Indeed, the $\alpha$-Rényi divergence is defined when $\sigma_\alpha^2 = \alpha\sigma_Q^2 + (1 - \alpha)\sigma_P^2 > 0$, i.e., $\alpha < \frac{\sigma_P^2}{\sigma_P^2 - \sigma_Q^2}$.

## C.2  Equal variances

If $\sigma_Q^2 = \sigma_P^2 = \sigma^2$, the importance weights have the following expression:

$$w(x) = \exp\left(\frac{\mu_P - \mu_Q}{\sigma^2}\left(x - \frac{\mu_P + \mu_Q}{2}\right)\right), \tag{17}$$

for $x \sim Q$. The weight $w(x)$ is clearly unbounded and has 0 as infimum value. Let us investigate its probability density function.

**Proposition C.3.** *Let $Q \sim \mathcal{N}(\mu_Q, \sigma^2)$ be the behavioral distribution and $P \sim \mathcal{N}(\mu_P, \sigma^2)$ be the target distribution. The probability density function of $w(x) = q(x)/p(x)$ is given by:*

$$f_w(y) = \frac{|\widetilde{\sigma}|}{\sqrt{2\pi}y^{\frac{3}{2}}}\exp\left(-\frac{1}{2}\left(\widetilde{\mu}^2 + \widetilde{\sigma}^2(\log y)^2\right)\right), \tag{18}$$

*where $\widetilde{\mu} = \frac{\mu_P - \mu_Q}{2\sigma}$ and $\widetilde{\sigma} = \frac{\sigma}{\mu_P - \mu_Q}$.*

*Proof.* We start computing the c.d.f.:

$$F_w(y) = \Pr\left(\exp\left\{\frac{\mu_P - \mu_Q}{\sigma^2}\left(x - \frac{\mu_P + \mu_Q}{2}\right)\right\} \leq y\right) = \Pr\left(\frac{\mu_P - \mu_Q}{\sigma^2}\left(x - \frac{\mu_P + \mu_Q}{2}\right) \leq \log y\right).$$

First, we consider the case $\mu_P - \mu_Q > 0$ and observe that $x = \mu_Q + \sigma z$, where $z \sim \mathcal{N}(0, 1)$:

$$F_w(y) = \Pr\left(x \leq \frac{\mu_P + \mu_Q}{2} + \frac{\sigma^2}{\mu_P - \mu_Q}\log y\right) = \Pr\left(z \leq \frac{\mu_P - \mu_Q}{2\sigma} + \frac{\sigma}{\mu_P - \mu_Q}\log y\right).$$

We call $\widetilde{\mu} = \frac{\mu_P - \mu_Q}{2\sigma}$ and $\widetilde{\sigma} = \frac{\sigma}{\mu_P - \mu_Q}$ and we have:

$$F_w(y) = \Pr\left(z \leq \widetilde{\mu} + \widetilde{\sigma}\log y\right) = \Phi\left(\widetilde{\mu} + \widetilde{\sigma}\log y\right).$$

We take the derivative in order to get the density function:

$$f_w(y) = \frac{\partial F_w(y)}{\partial y} = \frac{\widetilde{\sigma}}{y}\frac{1}{\sqrt{2\pi}}\exp\left(-\frac{1}{2}\left(\widetilde{\mu} + \widetilde{\sigma}\log y\right)^2\right) = \frac{\widetilde{\sigma}}{\sqrt{2\pi}y^{\widetilde{\mu}\widetilde{\sigma}+1}}\exp\left(-\frac{1}{2}\left(\widetilde{\mu}^2 + \widetilde{\sigma}^2(\log y)^2\right)\right).$$

For the case $\mu_P - \mu_Q < 0$ the derivation is symmetric and the p.d.f. differs only by a minus sign. We account for this fact by considering $|\widetilde{\sigma}|$ in the final formula. $\square$

In the case of equal variances, the tail behavior is different.

**Proposition C.4.** *If $\sigma_P^2 = \sigma_Q^2$ then for any $\alpha > 0$ there exist $c > 0$ and $y_0 > 0$ such that for any $y \geq y_0$, the p.d.f. can be upper bounded as $f_w(y) \leq cy^{-\alpha}$.*

(a) equal variance

(b) equal mean

Figure 3: Probability density function of the importance weights when the behavioral distribution is $\mathcal{N}(0,1)$ and the mean is changed keeping the variance equal to 1 (a) or the variance is changed keeping the target mean equal to 1 (b).

*Proof.* Condensing all the constants in $c$, the p.d.f. can be written as:

$$f_w(y) = cy^{-3/2} \exp\left((\log y)^2\right)^{-\frac{\tilde{\sigma}^2}{2}}. \tag{19}$$

For any $\alpha > 0$, let us solve the following inequality:

$$y^{3/2} \exp\left((\log y)^2\right)^{\frac{\tilde{\sigma}^2}{2}} \geq y^\alpha \quad \implies \quad y \geq \exp\left(\frac{2}{\tilde{\sigma}^2}\left(\alpha - \frac{3}{2}\right)\right). \tag{20}$$

Thus, for $y \geq \exp\left(\frac{2}{\tilde{\sigma}^2}\left(\alpha - \frac{3}{2}\right)\right)$ we have that $f_w(y) \leq cy^{-\alpha}$. $\square$

This is sufficient to ensure the existence of the moments of any order, indeed the corresponding Rényi divergence is: $\frac{\alpha(\mu_P - \mu_Q)^2}{2\sigma^2}$. By the way, the distribution of $w(x)$ remains subexponential, as $\exp\left((\log y)^2\right)^{-\frac{\tilde{\sigma}^2}{2}} \geq e^{-\eta y}$ for sufficiently large $y$.

Figure 3 reports the p.d.f. of the importance weights for different values of mean and variance of the target distribution.

# D  Analysis of the SN Estimator

In this Appendix, we provide some results regarding bias and variance of the self-normalized importance sampling estimator. Let us start with the following result, derived from [8], that bounds the expected squared difference between non-self-normalized weight $w(x)$ and self-normalized weight $\widetilde{w}(x)$.

**Lemma D.1.** *Let $P$ and $Q$ be two probability measures on the measurable space $(\mathcal{X}, \mathcal{F})$ such that $P \ll Q$ and $d_2(P\|Q) < +\infty$. Let $x_1, x_2, \ldots, x_N$ i.i.d. random variables sampled from $Q$. Then, for $N > 0$ and for any $i = 1, 2, \ldots, N$ it holds that:*

$$\mathbb{E}_{\mathbf{x}\sim Q}\left[\left(\widetilde{w}_{P/Q}(x_i) - \frac{w_{P/Q}(x_i)}{N}\right)^2\right] \leq \frac{d_2(P\|Q) - 1}{N}. \tag{21}$$

*Proof.* The result derives from simple algebraic manipulations and from the fact that $\mathbb{V}\mathrm{ar}_{x\sim Q}\left[w_{P/Q}(x)\right] = d_2(P\|Q) - 1$.

$$\mathbb{E}_{\mathbf{x}\sim Q}\left[\left(\widetilde{w}_{P/Q}(x_i) - \frac{w_{P/Q}(x_i)}{N}\right)^2\right] = \mathbb{E}_{\mathbf{x}\sim Q}\left[\left(\frac{w_{P/Q}(x_i)}{\sum_{j=1}^N w_{P/Q}(x_j)}\right)^2\left(1 - \frac{\sum_{j=1}^N w_{P/Q}(x_j)}{N}\right)^2\right]$$

$$\leq \mathbb{E}_{\mathbf{x}\sim Q}\left[\left(1 - \frac{\sum_{j=1}^N w_{P/Q}(x_j)}{N}\right)^2\right] = \mathbb{V}\mathrm{ar}_{\mathbf{x}\sim Q}\left[\frac{\sum_{j=1}^N w_{P/Q}(x_j)}{N}\right]$$

$$= \frac{1}{N} \operatorname*{Var}_{x_1 \sim Q} \left[ w_{P/Q}(x_1) \right] = \frac{d_2(P\|Q) - 1}{N}.$$

<div align="right">□</div>

A similar argument can be used to derive a bound on the bias of the SN estimator.

**Proposition D.1.** *Let $P$ and $Q$ be two probability measures on the measurable space $(\mathcal{X}, \mathcal{F})$ such that $P \ll Q$ and $d_2(P\|Q) < +\infty$. Let $x_1, x_2, \ldots, x_N$ i.i.d. random variables sampled from $Q$ and $f : \mathcal{X} \to \mathbb{R}$ be a bounded function ($\|f\|_\infty < \infty$). Then, the bias of the SN estimator can be bounded as:*

$$\left| \mathbb{E}_{\mathbf{x} \sim Q} \left[ \widetilde{\mu}_{P/Q} - \mathbb{E}_{x \sim P}[f(x)] \right] \right| \leq \|f\|_\infty \min \left\{ 2, \sqrt{\frac{d_2(P\|Q) - 1}{N}} \right\}. \tag{22}$$

*Proof.* Since it holds that $|\widetilde{\mu}_{P/Q}| \leq \|f\|_\infty$ the bias cannot be larger than $2\|f\|_\infty$. We now derive a bound for the bias that vanishes as $N \to \infty$. We exploit the fact that the IS estimator is unbiased, i.e., $\mathbb{E}_{\mathbf{x} \sim Q} \left[ \widehat{\mu}_{P/Q} \right] = \mathbb{E}_{x \sim P}[f(x)]$.

$$\left| \mathbb{E}_{\mathbf{x} \sim Q} \left[ \widetilde{\mu}_{P/Q} - \mathbb{E}_{x \sim P}[f(x)] \right] \right| = \left| \mathbb{E}_{\mathbf{x} \sim Q} \left[ \widetilde{\mu}_{P/Q} - \mathbb{E}_{\mathbf{x} \sim Q} \left[ \widehat{\mu}_{P/Q} \right] \right] \right| = \left| \mathbb{E}_{\mathbf{x} \sim Q} \left[ \widetilde{\mu}_{P/Q} - \widehat{\mu}_{P/Q} \right] \right|$$

$$\leq \mathbb{E}_{\mathbf{x} \sim Q} \left[ \left| \widetilde{\mu}_{P/Q} - \widehat{\mu}_{P/Q} \right| \right] =$$

$$= \mathbb{E}_{\mathbf{x} \sim Q} \left[ \left| \frac{\sum_{i=1}^N w_{P/Q}(x_i) f(x_i)}{\sum_{i=1}^N w_{P/Q}(x_i)} - \frac{\sum_{i=1}^N w_{P/Q}(x_i) f(x_i)}{N} \right| \right]$$

$$= \mathbb{E}_{\mathbf{x} \sim Q} \left[ \left| \frac{\sum_{i=1}^N w_{P/Q}(x_i) f(x_i)}{\sum_{i=1}^N w_{P/Q}(x_i)} \right| \left| 1 - \frac{\sum_{i=1}^N w_{P/Q}(x_i)}{N} \right| \right] \tag{23}$$

$$\leq \mathbb{E}_{\mathbf{x} \sim Q} \left[ \left( \frac{\sum_{i=1}^N w_{P/Q}(x_i) f(x_i)}{\sum_{i=1}^N w_{P/Q}(x_i)} \right)^2 \right]^{\frac{1}{2}} \mathbb{E}_{\mathbf{x} \sim Q} \left[ \left( 1 - \frac{\sum_{i=1}^N w_{P/Q}(x_i)}{N} \right)^2 \right]^{\frac{1}{2}} \tag{24}$$

$$\leq \|f\|_\infty \sqrt{\frac{d_2(P\|Q) - 1}{N}}, \tag{25}$$

where (24) follows from (23) by applying Cauchy-Schwartz inequality and (25) is obtained by observing that $\left( \frac{\sum_{i=1}^N w_{P/Q}(x_i) f(x_i)}{\sum_{i=1}^N w_{P/Q}(x_i)} \right)^2 \leq \|f\|_\infty^2$. <div align="right">□</div>

Bounding the variance of the SN estimator is non-trivial since the the normalization term makes all the samples interdependent. Exploiting the boundedness of $\widetilde{\mu}_{P/Q}$ we can derive trivial bounds like: $\operatorname{Var}_{\mathbf{x} \sim Q} \left[ \widetilde{\mu}_{P/Q} \right] \leq \|f\|_\infty^2$. However, this bound does not shrink with the number of samples $N$. Several approximations of the variance have been proposed, like the following derived using the delta method [60, 29]:

$$\operatorname*{Var}_{\mathbf{x} \sim Q} \left[ \widetilde{\mu}_{P/Q} \right] = \frac{1}{N} \mathbb{E}_{x_1 \sim Q} \left[ w_{P/Q}^2(x_1) \left( f(x_1) - \mathbb{E}_{x \sim P}[f(x)] \right)^2 \right] + o(N^{-2}). \tag{26}$$

We will not use the approximate expression for the variance, but we will directly bound the Mean Squared Error (MSE) of the SN estimator, which is the sum of the variance and the bias squared.

**Proposition D.2.** *Let $P$ and $Q$ be two probability measures on the measurable space $(\mathcal{X}, \mathcal{F})$ such that $P \ll Q$ and $d_2(P\|Q) < +\infty$. Let $x_1, x_2, \ldots, x_N$ i.i.d. random variables sampled from $Q$ and $f : \mathcal{X} \to \mathbb{R}$ be a bounded function ($\|f\|_\infty < +\infty$). Then, the MSE of the SN estimator can be bounded as:*

$$\operatorname{MSE}_{\mathbf{x} \sim Q} \left[ \widetilde{\mu}_{P/Q} \right] \leq 2\|f\|_\infty^2 \min \left\{ 2, \frac{2d_2(P\|Q) - 1}{N} \right\}. \tag{27}$$

*Proof.* First, recall that $\widetilde{\mu}_{P/Q}$ is bounded by $\|f\|_\infty$ thus its MSE cannot be larger than $4\|f\|_\infty^2$. The idea of the proof is to sum and subtract the IS estimator $\widehat{\mu}_{P/Q}$:

$$\operatorname{MSE}_{\mathbf{x} \sim Q} \left[ \widetilde{\mu}_{P/Q} \right] = \mathbb{E}_{\mathbf{x} \sim Q} \left[ \left( \widetilde{\mu}_{P/Q} - \mathbb{E}_{x \sim P}[f(x)] \right)^2 \right]$$

<div align="center">21</div>

$$= \mathop{\mathbb{E}}_{\mathbf{x}\sim Q}\left[\left(\widetilde{\mu}_{P/Q} - \mathop{\mathbb{E}}_{x\sim P}[f(x)] \pm \widehat{\mu}_{P/Q}\right)^2\right] \tag{28}$$

$$\leq 2\mathop{\mathbb{E}}_{\mathbf{x}\sim Q}\left[\left(\widetilde{\mu}_{P/Q} - \widehat{\mu}_{P/Q}\right)^2\right] + 2\mathop{\mathbb{E}}_{\mathbf{x}\sim Q}\left[\left(\widehat{\mu}_{P/Q} - \mathop{\mathbb{E}}_{x\sim P}[f(x)]\right)^2\right] \tag{29}$$

$$\leq 2\mathop{\mathbb{E}}_{\mathbf{x}\sim Q}\left[\left(\frac{\sum_{i=1}^N w_{P/Q}(x_i)f(x_i)}{\sum_{i=1}^N w_{P/Q}(x_i)}\right)^2\left(1 - \frac{\sum_{i=1}^N w_{P/Q}(x_i)}{N}\right)^2\right] + 2\mathop{\mathbb{V}\mathrm{ar}}_{\mathbf{x}\sim Q}\left[\widehat{\mu}_{P/Q}\right] \tag{30}$$

$$\leq 2\|f\|_\infty^2 \mathop{\mathbb{E}}_{\mathbf{x}\sim Q}\left[\left(1 - \frac{\sum_{i=1}^N w_{P/Q}(x_i)}{N}\right)^2\right] + 2\mathop{\mathbb{V}\mathrm{ar}}_{\mathbf{x}\sim Q}\left[\widehat{\mu}_{P/Q}\right] \tag{31}$$

$$\leq 2\|f\|_\infty^2 \mathop{\mathbb{V}\mathrm{ar}}_{\mathbf{x}\sim Q}\left[\frac{\sum_{i=1}^N w_{P/Q}(x_i)}{N}\right] + 2\mathop{\mathbb{V}\mathrm{ar}}_{\mathbf{x}\sim Q}\left[\widehat{\mu}_{P/Q}\right] \tag{32}$$

$$\leq 2\|f\|_\infty^2 \frac{d_2(P\|Q) - 1}{N} + 2\|f\|_\infty^2 \frac{d_2(P\|Q)}{N} = 2\|f\|_\infty^2 \frac{2d_2(P\|Q) - 1}{N}, \tag{33}$$

where line (29) follows from line (28) by applying the inequality $(a + b)^2 \leq 2(a^2 + b^2)$, (31) follows from (30) by observing that $\left(\frac{\sum_{i=1}^N w_{P/Q}(x_i)f(x_i)}{\sum_{i=1}^N w_{P/Q}(x_i)}\right)^2 \leq \|f\|_\infty^2$. $\qquad\square$

We can use this result to provide a high confidence bound for the SN estimator.

**Proposition D.3.** *Let $P$ and $Q$ be two probability measures on the measurable space $(\mathcal{X}, \mathcal{F})$ such that $P \ll Q$ and $d_2(P\|Q) < +\infty$. Let $x_1, x_2, \ldots, x_N$ i.i.d. random variables sampled from $Q$ and $f: \mathcal{X} \to \mathbb{R}$ be a bounded function ($\|f\|_\infty < +\infty$). Then, for any $0 < \delta \leq 1$ and $N > 0$ with probability at least $1 - \delta$:*

$$\mathop{\mathbb{E}}_{x\sim P}[f(x)] \geq \frac{1}{N}\sum_{i=1}^N \widetilde{w}_{P/Q}(x_i)f(x_i) - 2\|f\|_\infty \min\left\{1, \sqrt{\frac{d_2(P\|Q)(4 - 3\delta)}{\delta N}}\right\}.$$

*Proof.* The result is obtained by applying Cantelli's inequality and accounting for the bias. Consider the random variable $\widetilde{\mu}_{P/Q} = \frac{1}{N}\sum_{i=1}^N \widetilde{w}_{P/Q}(x_i)f(x_i)$ and let $\widetilde{\lambda} = \lambda - \left|\mathbb{E}_{x\sim P}[f(x)] - \mathbb{E}_{\mathbf{x}\sim P}[\widetilde{\mu}_{P/Q}]\right|$:

$$\Pr\left(\widetilde{\mu}_{P/Q} - \mathop{\mathbb{E}}_{x\sim P}[f(x)] \geq \lambda\right) = \Pr\left(\widetilde{\mu}_{P/Q} - \mathop{\mathbb{E}}_{\mathbf{x}\sim P}\left[\widetilde{\mu}_{P/Q}\right] \geq \lambda + \mathop{\mathbb{E}}_{x\sim P}[f(x)] - \mathop{\mathbb{E}}_{\mathbf{x}\sim P}\left[\widetilde{\mu}_{P/Q}\right]\right)$$

$$\leq \Pr\left(\widetilde{\mu}_{P/Q} - \mathop{\mathbb{E}}_{\mathbf{x}\sim P}\left[\widetilde{\mu}_{P/Q}\right] \geq \lambda - \left|\mathop{\mathbb{E}}_{x\sim P}[f(x)] - \mathop{\mathbb{E}}_{\mathbf{x}\sim P}\left[\widetilde{\mu}_{P/Q}\right]\right|\right)$$

$$= \Pr\left(\widetilde{\mu}_{P/Q} - \mathop{\mathbb{E}}_{\mathbf{x}\sim P}\left[\widetilde{\mu}_{P/Q}\right] \geq \widetilde{\lambda}\right).$$

Now we apply Cantelli's inequality:

$$\Pr\left(\widetilde{\mu}_{P/Q} - \mathop{\mathbb{E}}_{x\sim P}[f(x)] \geq \lambda\right) \leq \Pr\left(\widetilde{\mu}_{P/Q} - \mathop{\mathbb{E}}_{x\sim P}\left[\widetilde{\mu}_{P/Q}\right] \geq \widetilde{\lambda}\right) \leq \frac{1}{1 + \frac{\widetilde{\lambda}^2}{\mathbb{V}\mathrm{ar}_{\mathbf{x}\sim Q}[\widetilde{\mu}_{P/Q}]}}$$

$$= \frac{1}{1 + \frac{\left(\lambda - \left|\mathbb{E}_{x\sim P}[f(x)] - \mathbb{E}_{\mathbf{x}\sim P}[\widetilde{\mu}_{P/Q}]\right|\right)^2}{\mathbb{V}\mathrm{ar}_{\mathbf{x}\sim Q}[\widetilde{\mu}_{P/Q}]}}. \tag{34}$$

By calling $\delta = \frac{1}{1 + \frac{\left(\lambda - \left|\mathbb{E}_{x\sim P}[f(x)] - \mathbb{E}_{\mathbf{x}\sim P}[\widetilde{\mu}_{P/Q}]\right|\right)^2}{\mathbb{V}\mathrm{ar}_{\mathbf{x}\sim Q}[\widetilde{\mu}_{P/Q}]}}$ and considering the complementary event, we get that with probability at least $1 - \delta$ we have:

$$\mathop{\mathbb{E}}_{x\sim P}[f(x)] \geq \widetilde{\mu}_{P/Q} - \left|\mathop{\mathbb{E}}_{x\sim P}[f(x)] - \mathop{\mathbb{E}}_{\mathbf{x}\sim P}\left[\widetilde{\mu}_{P/Q}\right]\right| - \sqrt{\frac{1 - \delta}{\delta}\mathop{\mathbb{V}\mathrm{ar}}_{\mathbf{x}\sim Q}\left[\widetilde{\mu}_{P/Q}\right]} \tag{35}$$

Then we bound the bias term $\left|\mathbb{E}_{x\sim P}[f(x)] - \mathbb{E}_{\mathbf{x}\sim P}\left[\widetilde{\mu}_{P/Q}\right]\right|$ with equation (22) and the variance term with the MSE in equation (27). With some simple algebraic manipulation we have:

$$\mathop{\mathbb{E}}_{x\sim P}[f(x)] \geq \widetilde{\mu}_{P/Q} - \|f\|_\infty\sqrt{\frac{d_2(P\|Q) - 1}{N}} - \|f\|_\infty\sqrt{\frac{1 - \delta}{\delta}\frac{2(2d_2(P\|Q) - 1)}{N}}$$

$$\geq \widetilde{\mu}_{P/Q} - \|f\|_\infty\sqrt{\frac{d_2(P\|Q)}{N}} - \|f\|_\infty\sqrt{\frac{1 - \delta}{\delta}\frac{4d_2(P\|Q)}{N}}$$

$$= \widetilde{\mu}_{P/Q} - \|f\|_\infty \sqrt{\frac{d_2(P\|Q)}{N}} \left(1 + 2\sqrt{\frac{1-\delta}{\delta}}\right)$$

$$\geq \widetilde{\mu}_{P/Q} - 2\|f\|_\infty \sqrt{\frac{d_2(P\|Q)}{N}} \sqrt{1 + \frac{4(1-\delta)}{\delta}}$$

$$\geq \widetilde{\mu}_{P/Q} - 2\|f\|_\infty \sqrt{\frac{d_2(P\|Q)(4-3\delta)}{\delta N}},$$

where the last line follows from the fact that $\sqrt{a} + \sqrt{b} \leq 2\sqrt{a+b}$ for any $a, b \geq 0$. Finally, recalling that the range of the SN estimator is $2\|f\|_\infty$ we get the result. $\qquad \square$

It is worth noting that, apart for the constants, the bound has the same dependence on $d_2$ as in Theorem 4.1. Thus, by suitably redefining the hyperparameter $\lambda$ we can optimize the same surrogate objective function for both IS and SN estimators.

## E    Implementation details

In this Appendix, we provide some aspects about our implementation of POIS.

### E.1    Line Search

At each offline iteration $k$ the parameter update is performed in the direction of $\mathcal{G}(\boldsymbol{\theta}_k^j)^{-1}\nabla_{\boldsymbol{\theta}_k^j}\mathcal{L}(\boldsymbol{\theta}_k^j/\boldsymbol{\theta}_0^j)$ with a step size $\alpha_k$ determined in order to maximize the improvement. For brevity we will remove subscripts and dependence on $\boldsymbol{\theta}_0^j$ from the involved quantities. The rationale behind our line search is the following. Suppose that our objective function $\mathcal{L}(\boldsymbol{\theta})$, restricted to the gradient direction $\mathcal{G}^{-1}(\boldsymbol{\theta})\nabla_{\boldsymbol{\theta}}\mathcal{L}(\boldsymbol{\theta})$, represents a concave parabola in the Riemann manifold having $\mathcal{G}(\boldsymbol{\theta})$ as Riemann metric tensor. Suppose we know a point $\boldsymbol{\theta}_0$, the Riemann gradient in that point $\mathcal{G}(\boldsymbol{\theta}_0)^{-1}\nabla_{\boldsymbol{\theta}}\mathcal{L}(\boldsymbol{\theta}_0)$ and another point: $\boldsymbol{\theta}_l = \boldsymbol{\theta}_0 + \alpha_l\mathcal{G}(\boldsymbol{\theta}_0)^{-1}\nabla_{\boldsymbol{\theta}}\mathcal{L}(\boldsymbol{\theta}_0)$. For both points we know the value of the loss function: $\mathcal{L}_0 = \mathcal{L}(\boldsymbol{\theta}_0)$ and $\mathcal{L}_l = \mathcal{L}(\boldsymbol{\theta}_l)$ and indicate with $\Delta\mathcal{L}_l = \mathcal{L}_l - \mathcal{L}_0$ the objective function improvement. Having this information we can compute the vertex of that parabola, which is its global maximum. Let us call $l(\alpha) = \mathcal{L}\left(\boldsymbol{\theta}_0 + \alpha\mathcal{G}^{-1}(\boldsymbol{\theta}_0)\nabla_{\boldsymbol{\theta}}\mathcal{L}(\boldsymbol{\theta}_0)\right) - \mathcal{L}(\boldsymbol{\theta}_0)$, being a parabola it can be expressed as $l(\alpha) = a\alpha^2 + b\alpha + c$. Clearly, $c = 0$ by definition of $l(\alpha)$; $a$ and $b$ can be determined by enforcing the conditions:

$$b = \frac{\partial l}{\partial \alpha}\bigg|_{\alpha=0} = \frac{\partial}{\partial \alpha}\mathcal{L}\left(\boldsymbol{\theta}_0 + \alpha\mathcal{G}^{-1}(\boldsymbol{\theta}_0)\nabla_{\boldsymbol{\theta}}\mathcal{L}(\boldsymbol{\theta}_0)\right) - \mathcal{L}(\boldsymbol{\theta}_0)|_{\alpha=0} =$$

$$= \nabla_{\boldsymbol{\theta}}\mathcal{L}(\boldsymbol{\theta}_0)^T\mathcal{G}^{-1}(\boldsymbol{\theta}_0)\nabla_{\boldsymbol{\theta}}\mathcal{L}(\boldsymbol{\theta}_0) =$$

$$= \|\nabla_{\boldsymbol{\theta}}\mathcal{L}(\boldsymbol{\theta}_0)\|^2_{\mathcal{G}^{-1}(\boldsymbol{\theta}_0)},$$

$$l(\alpha_l) = a\alpha_l^2 + b\alpha_l = a\alpha_l^2 + \|\nabla_{\boldsymbol{\theta}}\mathcal{L}(\boldsymbol{\theta}_0)\|^2_{\mathcal{G}^{-1}(\boldsymbol{\theta}_0)}\alpha_l = \Delta\mathcal{L}_l \quad \Longrightarrow$$

$$\Longrightarrow \quad a = \frac{\Delta\mathcal{L}_l - \|\nabla_{\boldsymbol{\theta}}\mathcal{L}(\boldsymbol{\theta}_0)\|^2_{\mathcal{G}^{-1}(\boldsymbol{\theta}_0)}\alpha_l}{\alpha_l^2}.$$

Therefore, the parabola has the form:

$$l(\alpha) = \frac{\Delta\mathcal{L}_l - \|\nabla_{\boldsymbol{\theta}}\mathcal{L}(\boldsymbol{\theta}_0)\|^2_{\mathcal{G}^{-1}(\boldsymbol{\theta}_0)}\alpha_l}{\alpha_l^2}\alpha^2 + \|\nabla_{\boldsymbol{\theta}}\mathcal{L}(\boldsymbol{\theta}_0)\|^2_{\mathcal{G}^{-1}(\boldsymbol{\theta}_0)}\alpha. \tag{36}$$

Clearly, the parabola is concave only if $\Delta\mathcal{L}_l < \|\nabla_{\boldsymbol{\theta}}\mathcal{L}(\boldsymbol{\theta}_0)\|^2_{\mathcal{G}^{-1}(\boldsymbol{\theta}_0)}\alpha_l$. The vertex is located at:

$$\alpha_{l+1} = \frac{\|\nabla_{\boldsymbol{\theta}}\mathcal{L}(\boldsymbol{\theta}_0)\|^2_{\mathcal{G}^{-1}(\boldsymbol{\theta}_0)}\alpha_l^2}{2\left(\|\nabla_{\boldsymbol{\theta}}\mathcal{L}(\boldsymbol{\theta}_0)\|^2_{\mathcal{G}^{-1}(\boldsymbol{\theta}_0)}\alpha_l - \Delta\mathcal{L}_l\right)}. \tag{37}$$

To simplify the expression, like in [25] we define $\alpha_l = \epsilon_l/\|\nabla_{\boldsymbol{\theta}}\mathcal{L}(\boldsymbol{\theta}_0)\|^2_{\mathcal{G}^{-1}(\boldsymbol{\theta}_0)}$. Thus, we get:

$$\epsilon_{l+1} = \frac{\epsilon_l^2}{2(\epsilon_l - \Delta\mathcal{L}_l)}. \tag{38}$$

Of course, we need also to manage the case in which the parabola is convex, i.e., $\Delta\mathcal{L}_l \geq \|\nabla_{\boldsymbol{\theta}}\mathcal{L}(\boldsymbol{\theta}_0)\|^2_{\mathcal{G}^{-1}(\boldsymbol{\theta}_0)}\alpha_l$. Since our objective function is not really a parabola we reinterpret the two cases: i) $\Delta\mathcal{L}_l > \|\nabla_{\boldsymbol{\theta}}\mathcal{L}(\boldsymbol{\theta}_0)\|^2_{\mathcal{G}^{-1}(\boldsymbol{\theta}_0)}\alpha_l$, the function is sublinear and in this case we use (38) to determine the new step size $\alpha_{l+1} = \epsilon_{l+1}/\|\nabla_{\boldsymbol{\theta}}\mathcal{L}(\boldsymbol{\theta}_0)\|^2_{\mathcal{G}^{-1}(\boldsymbol{\theta}_0)}$; ii) $\Delta\mathcal{L}_l \geq \|\nabla_{\boldsymbol{\theta}}\mathcal{L}(\boldsymbol{\theta}_0)\|^2_{\mathcal{G}^{-1}(\boldsymbol{\theta}_0)}\alpha_l$, the function is superlinear, in this case we increase the step size multiplying by $\eta > 1$, i.e., $\alpha_{l+1} = \eta\alpha_l$. Finally the update rule becomes:

$$\epsilon_{l+1} = \begin{cases} \eta\epsilon_l & \text{if } \Delta\mathcal{L}_l > \frac{\epsilon_l(2\eta-1)}{2\eta} \\ \frac{\epsilon_l^2}{2(\epsilon_l - \Delta\mathcal{L}_l)} & \text{otherwise} \end{cases}. \tag{39}$$

The procedure is iterated until a maximum number of attempts is reached (say 30) or the objective function improvement is too small (say 1e-4). The pseudocode of the line search is reported in Algorithm 3.

---

**Algorithm 3** Parabolic Line Search

    **Input**: $\text{tol}_{\Delta\mathcal{L}} = 1e-4$, $M_{\text{ls}} = 30$, $\mathcal{L}_0$
    **Output** : $\alpha^*$
  $\alpha_0 = 0$
  $\epsilon_1 = 1$
  $\Delta\mathcal{L}_{k-1} = -\infty$
  **for** $l = 1, 2, \ldots, M_{\text{ls}}$ **do**
    $\alpha_l = \epsilon_l/\|\nabla_{\boldsymbol{\theta}}\mathcal{L}(\boldsymbol{\theta}_0)\|^2_{\mathcal{G}^{-1}(\boldsymbol{\theta}_0)}$
    $\boldsymbol{\theta}_l = \alpha_l \mathcal{G}^{-1}(\boldsymbol{\theta}_0)\nabla_{\boldsymbol{\theta}}\mathcal{L}(\boldsymbol{\theta}_0)$
    $\Delta\mathcal{L}_l = \mathcal{L}_l - \mathcal{L}_0$
    **if** $\Delta\mathcal{L}_l < \Delta\mathcal{L}_{l-1} + \text{tol}_{\Delta\mathcal{L}}$ **then**
      **return** $\alpha_{l-1}$
    **end if**
    $\epsilon_{l+1} = \begin{cases} \eta\epsilon_l & \text{if } \Delta\mathcal{L}_l > \frac{\epsilon_l(1-2\eta)}{2\eta} \\ \frac{\epsilon_l^2}{2(\epsilon_l - \Delta\mathcal{L}_l)} & \text{otherwise} \end{cases}$
  **end for**

---

### E.2 Estimation of the Rényi divergence

In A-POIS, the Rényi divergence needs to be computed between the behavioral $p(\cdot|\boldsymbol{\theta})$ and target $p(\cdot|\boldsymbol{\theta}')$ distributions on trajectories. This is likely impractical as it requires to integrate over the trajectory space. Moreover, for stochastic environments it cannot be computed unless we know the transition model $P$. The following result provides an exact, although loose, bound to this quantity in the case of finite-horizon tasks.

**Proposition E.1.** *Let $p(\cdot|\boldsymbol{\theta})$ and $p(\cdot|\boldsymbol{\theta}')$ be the behavioral and target trajectory probability density functions. Let $H < \infty$ be the task-horizon. Then, it holds that:*

$$d_\alpha\left(p(\cdot|\boldsymbol{\theta}')\|p(\cdot|\boldsymbol{\theta})\right) \leq \left(\sup_{s \in \mathcal{S}} d_\alpha\left(\pi_{\boldsymbol{\theta}'}(\cdot|s)\|\pi_{\boldsymbol{\theta}}(\cdot|s)\right)\right)^H.$$

*Proof.* We prove the proposition by induction on the horizon $H$. We define $d_{\alpha,H}$ as the $\alpha$-Rényi divergence at horizon $H$. For $H = 1$ we have:

$$
\begin{aligned}
d_{\alpha,1}\left(p(\cdot|\boldsymbol{\theta}')\|p(\cdot|\boldsymbol{\theta})\right) &= \int_{\mathcal{S}} D(s_0) \int_{\mathcal{A}} \pi_{\boldsymbol{\theta}}(a_0|s_0) \left(\frac{\pi_{\boldsymbol{\theta}'}(a_0|s_0)}{\pi_{\boldsymbol{\theta}}(a_0|s_0)}\right)^\alpha \int_{\mathcal{S}} P(s_1|s_0, a_0)\, \mathrm{d}s_1\, \mathrm{d}a_0\, \mathrm{d}s_0 \\
&= \int_{\mathcal{S}} D(s_0) \int_{\mathcal{A}} \pi_{\boldsymbol{\theta}}(a_0|s_0) \left(\frac{\pi_{\boldsymbol{\theta}'}(a_0|s_0)}{\pi_{\boldsymbol{\theta}}(a_0|s_0)}\right)^\alpha \mathrm{d}a_0\, \mathrm{d}s_0 \\
&\leq \int_{\mathcal{S}} D(s_0)\, \mathrm{d}s_0 \sup_{s \in \mathcal{S}} \int_{\mathcal{A}} \pi_{\boldsymbol{\theta}}(a_0|s) \left(\frac{\pi_{\boldsymbol{\theta}'}(a_0|s)}{\pi_{\boldsymbol{\theta}}(a_0|s)}\right)^\alpha \mathrm{d}a_0 \\
&\leq \sup_{s \in \mathcal{S}} d_\alpha\left(\pi_{\boldsymbol{\theta}'}(\cdot|s)\|\pi_{\boldsymbol{\theta}}(\cdot|s)\right),
\end{aligned}
$$

where the last but one passage follows from Holder's inequality. Suppose that the proposition holds for any $H' < H$, let us prove the proposition for $H$.

$$d_{\alpha,H}\left(p(\cdot|\boldsymbol{\theta}')\|p(\cdot|\boldsymbol{\theta})\right) = \int_{\mathcal{S}} D(s_0) \cdots \int_{\mathcal{A}} \pi_{\boldsymbol{\theta}}(a_{H-2}|s_{H-2}) \left(\frac{\pi_{\boldsymbol{\theta}'}(a_{H-2}|s_{H-2})}{\pi_{\boldsymbol{\theta}}(a_{H-2}|s_{H-2})}\right)^{\alpha} \int_{\mathcal{S}} P(s_{H-1}|s_{H-2}, a_{H-2})$$

$$\times \int_{\mathcal{A}} \pi_{\boldsymbol{\theta}}(a_{H-1}|s_{H-1}) \left(\frac{\pi_{\boldsymbol{\theta}'}(a_{H-1}|s_{H-1})}{\pi_{\boldsymbol{\theta}}(a_{H-1}|s_{H-1})}\right)^{\alpha} \int_{\mathcal{S}} P(s_H|s_{H-1}, a_{H-1}) \, \mathrm{d}s_0 \ldots \mathrm{d}s_{H-1}$$

$$\times \mathrm{d}a_{H-2} \, \mathrm{d}s_{H-1} \, \mathrm{d}a_{H-1} \, \mathrm{d}s_H$$

$$= \int_{\mathcal{S}} D(s_0) \cdots \int_{\mathcal{A}} \pi_{\boldsymbol{\theta}}(a_{H-2}|s_{H-2}) \left(\frac{\pi_{\boldsymbol{\theta}'}(a_{H-2}|s_{H-2})}{\pi_{\boldsymbol{\theta}}(a_{H-2}|s_{H-2})}\right)^{\alpha} \int_{\mathcal{S}} P(s_{H-1}|s_{H-2}, a_{H-2})$$

$$\times \int_{\mathcal{A}} \pi_{\boldsymbol{\theta}}(a_{H-1}|s_{H-1}) \left(\frac{\pi_{\boldsymbol{\theta}'}(a_{H-1}|s_{H-1})}{\pi_{\boldsymbol{\theta}}(a_{H-1}|s_{H-1})}\right)^{\alpha} \mathrm{d}s_0 \ldots \mathrm{d}s_{H-1} \, \mathrm{d}a_{H-2} \, \mathrm{d}s_{H-1} \, \mathrm{d}a_{H-1}$$

$$\leq \int_{\mathcal{S}} D(s_0) \cdots \int_{\mathcal{A}} \pi_{\boldsymbol{\theta}}(a_{H-2}|s_{H-2}) \left(\frac{\pi_{\boldsymbol{\theta}'}(a_{H-2}|s_{H-2})}{\pi_{\boldsymbol{\theta}}(a_{H-2}|s_{H-2})}\right)^{\alpha} \int_{\mathcal{S}} P(s_{H-1}|s_{H-2}, a_{H-2})$$

$$\times \mathrm{d}s_0 \ldots \mathrm{d}s_{H-1} \, \mathrm{d}a_{H-2} \, \mathrm{d}s_{H-1} \times \sup_{s \in \mathcal{S}} \int_{\mathcal{A}} \pi_{\boldsymbol{\theta}}(a_{H-1}|s) \left(\frac{\pi_{\boldsymbol{\theta}'}(a_{H-1}|s)}{\pi_{\boldsymbol{\theta}}(a_{H-1}|s)}\right)^{\alpha} \mathrm{d}a_{H-1}$$

$$\leq d_{\alpha,H-1}\left(p(\cdot|\boldsymbol{\theta}')\|p(\cdot|\boldsymbol{\theta})\right) \sup_{s \in \mathcal{S}} d_{\alpha}\left(\pi_{\boldsymbol{\theta}'}(\cdot|s)\|\pi_{\boldsymbol{\theta}}(\cdot|s)\right)$$

$$\leq \left(\sup_{s \in \mathcal{S}} d_{\alpha}\left(\pi_{\boldsymbol{\theta}'}(\cdot|s)\|\pi_{\boldsymbol{\theta}}(\cdot|s)\right)\right)^{H},$$

where we applied Holder's inequality again and the last passage is obtained for the inductive hypothesis. $\qquad\square$

The proposed bound, however, is typically ultraconservative, thus we propose two alternative estimators of the $\alpha$-Rényi divergence. The first estimator is obtained by simply rephrasing the definition (4) into a sample-based version:

$$\widehat{d}_{\alpha}\left(P\|Q\right) = \frac{1}{N} \sum_{i=1}^{N} \left(\frac{p(x_i)}{q(x_i)}\right)^{\alpha} = \frac{1}{N} \sum_{i=1}^{N} w_{P/Q}^{\alpha}(x_i), \tag{40}$$

where $x_i \sim Q$. This estimator is clearly unbiased and applies to any pair of probability distributions. However, in A-POIS $P$ and $Q$ are distributions over trajectories, their densities are expressed as products, thus the $\alpha$-Rényi divergence becomes:

$$d_{\alpha}\left(p(\cdot|\boldsymbol{\theta}')\|p(\cdot|\boldsymbol{\theta})\right) = \int_{\mathcal{T}} p(\cdot|\boldsymbol{\theta})(\tau) \left(\frac{p(\tau|\boldsymbol{\theta}')}{p(\tau|\boldsymbol{\theta})}\right)^{\alpha} \mathrm{d}\tau =$$

$$= \int_{\mathcal{T}} D(s_{\tau,0}) \prod_{t=0}^{H-1} P(s_{\tau,t+1}|s_{\tau,t}, a_{\tau,t}) \prod_{t=0}^{H-1} \pi_{\boldsymbol{\theta}}(a_{\tau,t}|s_{\tau,t}) \left(\frac{\pi_{\boldsymbol{\theta}'}(a_{\tau,t}|s_{\tau,t})}{\pi_{\boldsymbol{\theta}}(a_{\tau,t}|s_{\tau,t})}\right)^{\alpha} \mathrm{d}\tau.$$

Since both $\pi_{\boldsymbol{\theta}}$ and $\pi_{\boldsymbol{\theta}'}$ are known we are able to compute exactly for each state $d_{\alpha}\left(\pi_{\boldsymbol{\theta}'}(\cdot|s)\|\pi_{\boldsymbol{\theta}}(\cdot|s)\right)$ with no need to sample the action $a$. Therefore, we suggest to estimate the Rényi divergence between two trajectory distributions as:

$$\widehat{d}_{\alpha}\left(p(\cdot|\boldsymbol{\theta}')\|p(\cdot|\boldsymbol{\theta})\right) = \frac{1}{N} \sum_{i=1}^{N} \prod_{t=0}^{H-1} d_{\alpha}\left(\pi_{\boldsymbol{\theta}'}(\cdot|s_{\tau_i,t})\|\pi_{\boldsymbol{\theta}}(\cdot|s_{\tau_i,t})\right). \tag{41}$$

### E.3 Computation of the Fisher Matrix

In A-POIS the Fisher Information Matrix needs to be estimated off-policy from samples. We can use, for this purpose, the IS estimator:

$$\widehat{\mathcal{F}}(\boldsymbol{\theta}'/\boldsymbol{\theta}) = \frac{1}{N} \sum_{i=1}^{N} w_{\boldsymbol{\theta}'/\boldsymbol{\theta}}(\tau_i) \left(\sum_{t=0}^{H-1} \nabla_{\boldsymbol{\theta}'} \log \pi_{\boldsymbol{\theta}'}(a_{\tau_i,t}|s_{\tau_i,t})\right)^{T} \left(\sum_{t=0}^{H-1} \nabla_{\boldsymbol{\theta}'} \log \pi_{\boldsymbol{\theta}'}(a_{\tau_i,t}|s_{\tau_i,t})\right).$$

The SN estimator is obtained by replacing $w_{\boldsymbol{\theta}'/\boldsymbol{\theta}}(\tau_i)$ with $\widetilde{w}_{\boldsymbol{\theta}'/\boldsymbol{\theta}}(\tau_i)$. Those estimators become very unreliable when $\boldsymbol{\theta}'$ is far from $\boldsymbol{\theta}$, making them difficult to use in practice. On the contrary, in P-POIS

in presence of Gaussian hyperpolicies the FIM can be computed exactly [49]. If the hyperpolicy has diagonal covariance matrix, i.e., $\nu_{\boldsymbol{\mu},\boldsymbol{\sigma}} = \mathcal{N}(\boldsymbol{\mu}, \text{diag}(\boldsymbol{\sigma}^2))$, the FIM is also diagonal:

$$\mathcal{F}(\boldsymbol{\mu},\boldsymbol{\sigma}) = \left( \begin{array}{c|c} \text{diag}(1/\boldsymbol{\sigma}^2) & \mathbf{0} \\ \hline \mathbf{0} & 2\mathbf{I} \end{array} \right),$$

where $\mathbf{I}$ is a properly-sized identity matrix.

### E.4    Practical surrogate objective functions

In practice, the Rényi divergence term $d_2$ in the surrogate objective functions presented so far, either exact in P-POIS or approximate in A-POIS, tends to be overly-conservative. To mitigate this problem, by observing that $d_2(P\|Q)/N = 1/\text{ESS}(P\|Q)$ from equation (6) we can replace the whole quantity with an estimator like $\widehat{\text{ESS}}(P\|Q)$, as presented in equation (6). This leads to the following approximated surrogate objective functions:

$$\widetilde{\mathcal{L}}_\lambda^{\text{A}-\text{POIS}}(\boldsymbol{\theta'}/\boldsymbol{\theta}) = \frac{1}{N}\sum_{i=1}^{N} w_{\boldsymbol{\theta'}/\boldsymbol{\theta}}(\tau_i)R(\tau_i) - \frac{\lambda}{\sqrt{\widehat{\text{ESS}}\left(p(\cdot|\boldsymbol{\theta'})\|p(\cdot|\boldsymbol{\theta})\right)}},$$

$$\widetilde{\mathcal{L}}_\lambda^{\text{P}-\text{POIS}}(\boldsymbol{\rho'}/\boldsymbol{\rho}) = \frac{1}{N}\sum_{i=1}^{N} w_{\boldsymbol{\rho'}/\boldsymbol{\rho}}(\boldsymbol{\theta}_i)R(\tau_i) - \frac{\lambda}{\sqrt{\widehat{\text{ESS}}\left(\nu_{\boldsymbol{\rho'}}\|\nu_{\boldsymbol{\rho}}\right)}}.$$

Moreover, in all the experiments, we use the empirical maximum reward in place of the true $R_{\max}$.

### E.5    Practical P-POIS for Deep Neural Policies (N-POIS)

As mentioned in Section 6.2, P-POIS applied to deep neural policies suffers from a curse of dimensionality due to the high number of (scalar) parameters (which are $\sim 10^3$ for the network used in the experiments). The corresponding hyperpolicy is a multi-variate Gaussian (diagonal covariance) with a very high dimensionality. As a result, the Rényi divergence, used as a penalty, is extremely sensitive even to small perturbations, causing an overly-conservative behavior. First, we give up the exact Rényi computation and use the practical surrogate objective function $\widetilde{\mathcal{L}}_\lambda^{\text{P}-\text{POIS}}$ proposed in Appendix E.4. This, however, is not enough. The importance weights, being the products of thousands of probability densities, can easily become zero, preventing any learning. Hence, we decide to group the policy parameters in smaller blocks, and independently learn the corresponding hyperparameters. In general, we can define a family of $M$ orthogonal policy-parameter subspaces $\{\Theta_m \leq \Theta\}_{m=1}^M$, where $V \leq W$ reads "$V$ is a subspace of $W$". For each $\Theta_m$, we consider a multi-variate diagonal-covariance Gaussian with $\Theta_m$ as support, obtaining a corresponding hyperparameter subspace $\mathcal{P}_m \leq \mathcal{P}$. Then, for each $\mathcal{P}_m$, we compute a separate surrogate objective (where we employ self-normalized importance weights):

$$\widetilde{\mathcal{L}}_\lambda^{\text{N}-\text{POIS}}(\boldsymbol{\rho'}_m/\boldsymbol{\rho}_m) = \frac{1}{N}\sum_{i=1}^{N} \widetilde{w}_{\boldsymbol{\rho'}_m/\boldsymbol{\rho}_m}(\boldsymbol{\theta}_m^i)R(\tau_i) - \frac{\lambda}{\sqrt{\widehat{\text{ESS}}\left(\nu_{\boldsymbol{\rho'}_m}\|\nu_{\boldsymbol{\rho}_m}\right)}},$$

where $\boldsymbol{\rho}_m, \boldsymbol{\rho'}_m \in \mathcal{P}_m, \boldsymbol{\theta}_m \in \Theta_m$. Each objective is independently optimized via natural gradient ascent, where the step size is found via a line search as usual. It remains to define a meaningful grouping for the policy parameters, i.e., for the weights of the deep neural policy. We choose to group them by network unit, or neuron (counting output units but not input units). More precisely, let denote a network unit as a function:

$$U_i(\mathbf{x}|\boldsymbol{\theta}_m) = g(\mathbf{x}^T\boldsymbol{\theta}_m),$$

where $\mathbf{x}$ is the vector of the inputs to the unit (including a 1 that multiplies the bias parameter) and $g(\cdot)$ is an activation function. To each unit $U_m$ we associate a block $\Theta_m$ such that $\boldsymbol{\theta}_m \in \Theta_m$. In more connectivist-friendly terms, we group connections by the neuron they go into. For the network we used in the experiments, this reduces the order of the multivariate Gaussian hyperpolicies from $\sim 10^3$ to $\sim 10^2$. We call this practical variant of our algorithm Neuron-Based POIS (N-POIS). Although some design choices seem rather arbitrary, and independently optimizing hyperparameter blocks clearly neglects some potentially meaningful interactions, the practical results of N-POIS are promising, as reported in Section 6.2. Figure 4 is an ablation study showing the performance of P-POIS variants on Cartpole. Only using both the tricks discussed in this section, we are able to solve the task (this experiment is on 50 iterations only).

Figure 4: Ablation study for N-POIS (5 runs, 95% c.i.).

## F    Experiments Details

In this Appendix, we report the hyperparameter values used in the experimental evaluation and some additional plots and experiments. We adopted different criteria to decide the batch size: for linear policies at each iteration 100 episodes are collected regardless of their length, whereas for deep neural policies, in order to be fully comparable with [12], 50000 timesteps are collected at each iteration regardless of the resulting number of episodes (the last episode is cut so that the number of timesteps sums up exactly to 50000). Clearly, this difference is relevant only for episodic tasks.

### F.1    Linear policies

In the following we report the hyperparameters shared by all tasks and algorithms for the experiments with linear policies:

- Policy architecture: Normal distribution $\mathcal{N}(u_{\mathbf{M}}(\mathbf{s}), e^{2\mathbf{\Omega}})$, where the mean $u_{\mathbf{M}}(\mathbf{s}) = \mathbf{Ms}$ is a linear function in the state variables with no bias, and the variance is state-independent and parametrized as $e^{2\mathbf{\Omega}}$, with diagonal $\mathbf{\Omega}$.

- Number of runs: 20 (95% c.i.)

- seeds: 10, 109, 904, 160, 570, 662, 963, 100, 746, 236, 247, 689, 153, 947, 307, 42, 950, 315, 545, 178

- Policy initialization: mean parameters sampled from $\mathcal{N}(0, 0.01^2)$, variance initialized to 1

- Task horizon: 500

- Number of iterations: 500

- Maximum number of line search attempts (POIS only): 30

- Maximum number of offline iterations (POIS only): 10

- Episodes per iteration: 100

- Importance weight estimator (POIS only): IS for A-POIS, SN for P-POIS

- Natural gradient (POIS only): No for A-POIS, Yes for P-POIS

Table 3 reports the hyperparameters that have been tuned specifically for each task selecting the best combination based on the runs corresponding to the first 5 seeds.

Table 3: Task-specific hyperparameters for the experiments with linear policy. $\delta$ is the significance level for POIS while $\delta$ is the step-size for TRPO and PPO. In **bold**, the best hyperparameters found.

| Environment | A-POIS ($\delta$) | P-POIS ($\delta$) |
|---|---|---|
| Cart-Pole Balancing | 0.1, 0.2, 0.3, **0.4**, 0.5 | 0.1, 0.2, 0.3, **0.4**, 0.5, 0.6, 0.7, 0.8, 0.9 1 |
| Inverted Pendulum | 0.8, **0.9**, 0.99, 1 | 0.1, 0.2, 0.3, 0.4, 0.5, 0.6, 0.7, **0.8**, 0.9 1 |
| Mountain Car | 0.8, **0.9**, 0.99, 1 | 0.1, 0.2, 0.3, 0.4, 0.5, 0.6, 0.7, 0.8, 0.9, **1** |
| Acrobot | 0.1, 0.3, 0.5, **0.7**, 0.9 | 0.1, **0.2**, 0.3, 0.4, 0.5, 0.6, 0.7, 0.8, 0.9 1 |
| Double Inverted Pendulum | **0.1**, 0.2, 0.3, 0.4, 0.5 | **0.1**, 0.2, 0.3, 0.4, 0.5, 0.6, 0.7, 0.8, 0.9 1 |

| Environment | TRPO ($\delta$) | PPO ($\delta$) |
|---|---|---|
| Cart-Pole Balancing | 0.001, 0.01, **0.1**, 1 | 0.001, **0.01**, 0.1 , 1 |
| Inverted Pendulum | 0.001, **0.01**, 0.1, 1 | 0.001, **0.01**, 0.1, 1 |
| Mountain Car | 0.001, **0.01**, 0.1, 1 | 0.001, 0.01, 0.1, **1** |
| Acrobot | 0.001, 0.01, 0.1, **1** | 0.001, 0.01, 0.1, **1** |
| Double Inverted Pendulum | 0.001, 0.01, **0.1**, 1 | 0.001, 0.01, 0.1, **1** |

## F.2 Deep neural policies

In the following we report the hyperparameters shared by all tasks and algorithms for the experiments with deep neural policies:

- Policy architecture: Normal distribution $\mathcal{N}(u_{\mathbf{M}}(\mathbf{s}), e^{2\mathbf{\Omega}})$, where the mean $u_{\mathbf{M}}(\mathbf{s})$ is a 3-layers MLP (100, 50, 25) with bias (activation functions: tanh for hidden-layers, linear for output layer), the variance is state-independent and parametrized as $e^{2\mathbf{\Omega}}$ with diagonal $\mathbf{\Omega}$.
- Number of runs: 5 (95% c.i.)
- seeds: 10, 109, 904, 160, 570
- Policy initialization: uniform Xavier initialization [13], variance initialized to 1
- Task horizon: 500
- Number of iterations: 500
- Maximum number of line search attempts (POIS only): 30
- Maximum number of offline iterations (POIS only): 20
- Timesteps per iteration: 50000
- Importance weight estimator (POIS only): IS for A-POIS, SN for P-POIS
- Natural gradient (POIS only): No for A-POIS, Yes for P-POIS

Table 4 reports the hyperparameters that have been tuned specifically for each task selecting the best combination based on the runs corresponding to the 5 seeds.

Table 4: Task-specific hyperparameters for the experiments with deep neural policies. $\delta$ is the significance level for POIS. In **bold**, the best hyperparameters found.

| Environment | A-POIS ($\delta$) | P-POIS ($\delta$) |
|---|---|---|
| Cart-Pole Balancing | 0.9, **0.99**, 0.999 | 0.4, 0.5, **0.6**, 0.7, 0.8 |
| Mountain Car | 0.9, **0.99**, 0.999 | 0.1, 0.2, **0.3**, 0.4, 0.5, 0.6, 0.7, 0.8 |
| Double Inverted Pendulum | 0.9, **0.99**, 0.999 | 0.4, 0.5, 0.6, 0.7, **0.8** |
| Swimmer | 0.9, **0.99**, 0.999 | 0.4, 0.5, **0.6**, 0.7, 0.8 |

## F.3 Full experimental results

In this section, we report the complete set of results we obtained by testing the two versions of POIS.

In Figure 5 we report additional plots w.r.t. Figure 2 for A-POIS when changing the $\delta$ parameter in the Cartpole environment. It is worth noting that the value of $\delta$ has also an effect on the speed with which

the variance of the policy approaches zero. Indeed, smaller policy variances induce a larger Rényi divergence and thus with a higher penalization (small $\delta$) reducing the policy variance is discouraged. Moreover, we can see the values of the bound before and after the optimization. Clearly, the higher the value of $\delta$, the higher the value of the bound after the optimization process, as the penalization term is weaker. It is interesting to notice that when $\delta = 1$ the bound after the optimization reaches values that are impossible to reach for any policy and this is a consequence of the high uncertainty in the importance sampling estimator.

Figure 5: Standard Deviation of the policy ($\sigma$), value of the bound before and after the optimization as a function of the number of trajectories for A-POIS in the Cartpole environment for different values of $\delta$ (5 runs, 95% c.i.).

We report the comparative table taken from [12] containing all the benchmarked algorithms and the two versions of POIS (Table 5).

Table 5: Cumulative return compared with [12] on deep neural policies (5 runs, 95% c.i.). In **bold**, the performances that are not statistically significantly different from the best algorithm in each task.

| Algorithm | Cart-Pole Balancing | Mountain Car | Double Inverted Pendulum | Swimmer |
|---|---|---|---|---|
| Random | $77.1 \pm 0.0$ | $-415.4 \pm 0.0$ | $149.7 \pm 0.1$ | $-1.7 \pm 0.1$ |
| REINFORCE | $4693.7 \pm 14.0$ | $-67.1 \pm 1.0$ | $4116.5 \pm 65.2$ | $92.3 \pm 0.1$ |
| TNPG | $\mathbf{3986.4 \pm 748.9}$ | $\mathbf{-66.5 \pm 4.5}$ | $\mathbf{4455.4 \pm 37.6}$ | $\mathbf{96.0 \pm 0.2}$ |
| RWR | $\mathbf{4861.5 \pm 12.3}$ | $-79.4 \pm 1.1$ | $3614.8 \pm 368.1$ | $60.7 \pm 5.5$ |
| REPS | $565.6 \pm 137.6$ | $-275.6 \pm 166.3$ | $446.7 \pm 114.8$ | $3.8 \pm 3.3$ |
| TRPO | $\mathbf{4869.8 \pm 37.6}$ | $\mathbf{-61.7 \pm 0.9}$ | $\mathbf{4412.4 \pm 50.4}$ | $\mathbf{96.0 \pm 0.2}$ |
| DDPG | $4634.4 \pm 87.6$ | $-288.4 \pm 170.3$ | $2863.4 \pm 154.0$ | $85.8 \pm 1.8$ |
| A-POIS | $\mathbf{4842.8 \pm 13.0}$ | $-63.7 \pm 0.5$ | $\mathbf{4232.1 \pm 189.5}$ | $88.7 \pm 0.55$ |
| CEM | $4815.4 \pm 4.8$ | $-66.0 \pm 2.4$ | $2566.2 \pm 178.9$ | $68.8 \pm 2.4$ |
| CMA-ES | $2440.4 \pm 568.3$ | $-85.0 \pm 7.7$ | $1576.1 \pm 51.3$ | $64.9 \pm 1.4$ |
| P-POIS | $4428.1 \pm 138.6$ | $-78.9 \pm 2.5$ | $3161.4 \pm 959.2$ | $76.8 \pm 1.6$ |

In the following (Figure 6) we show the learning curves of POIS in its two versions for the experiments with deep neural policies.

(a) Inverted Double Pendulum

(b) Cartpole

(c) Mountain Car

(d) Swimmer

Figure 6: Average return as a function of the number of trajectories for A-POIS, P-POIS with deep neural policies (5 runs, 95% c.i.).