[Reviews · NeurIPS 2018]

Reviewer 1



Summary ======= The authors present a reinforcement learning technique based on importance sampling. A theoretical analysis is performed that shows how the importance sampling approach affect the upper bound of the expected performance of the target policy using samples from a behavioral policy. The authors propose a surrogate objective function that explicitly mitigates the variance of the policy update due to IS. Two algorithms are proposed based on natural gradients for control-based (learning low-level policy) and parameter-based problems (discrete low-level controller with stochastic upper-level policy). The algorithms were tested on standard control tasks and are compared to state of the art methods. Major comments ============= The authors investigate a rather general problem in modern reinforcement learning, that is, how we can efficiently reuse previous samples for off-policy reinforcement learning methods. The importance sampling (and similar) approach has been often used in the literature, but the actual impact on policy update variance have not been explicitly addressed in a (surrogate) objective function. The authors give a thorough theoretical investigation and provide two efficient algorithms for control and parameter based learning problems. The clarity is superb, the paper is easy to follow, while some details are difficult to understand without reading the supplementary material. I particularly liked the presentation of the related work from the aspect of control, or parameter based policy learning. The evaluation of the results is also impressive and the authors do not shy away of pointing out shortcomings of the proposed algorithms. Another aspect would be interesting to highlight is scalability w.r.t. the problem dimensionality. The tested methods are rather low dimensional, but sufficient for comparison to related methods. Minor comments ============= Is the assumption on constraining the reward between Rmin and Rmax really necessary? Does it have to be known explicitly? Response to Author Feedback ======================= The questions and suggestions have been properly addressed in the author feedback.

Reviewer 2



[UPDATE: I have read the author responses and I am satisfied with them] This paper proposes a novel importance sampling policy search method which accounts for the problem of large variance (low effective sample size) if one just does naive IS and the target distribution differs a lot from the behavior distribution. The authors take a more conservative approach, deriving an update rule based on natural gradient and considering a surrogate which is derived analytically. This is in essence the principle of most of the state-of-the-art algorithms for policy search. They develop both control-based and parameter-based formulations and provide a detailed comparison with some state-of-the-art policy optimization methods. This is a very well written paper and a solid piece of work. There are several nice contributions in this paper: from a theoretical side, the authors bring to the policy optimization field a set of useful tools from sampling and stochastic calculus which I don't think they are well established in the community. Moreover, they extensively review a vast piece of existing research done in the area. In addition, the proposed algorithms have comparable performance to the state-of-the-art methods. I have the following comments/questions to the authors: 1- line 185: "unacceptable limitations (...) target variance larger than the behavioral one" and line 109: "we require \sigma_P < 2\sigma_Q". -1st question: Is this not a contradiction? -2nd why would one use a target variance larger than the behavioral one? Usually, the behavioral one is more explorative (larger variance) and not the other way around. -3rd can you relate this with the experiments? 2- While it seems that authors consider general stochastic systems (line 100), they only focus on deterministic systems. In that sense, for example, (line 237) is misleading, since for deterministic systems the probability of a trajectory can be calculated without knowing the model. Isn't it? I strongly suggest to state from the beginning whether they consider deterministic dynamical systems or not. 3 - The numerical experiments are a bit disconnected from the theoretical parts of the paper. For example, given the strong emphasis that is made on the Effective Sample Size, I was expecting some numerical demonstration that showed the increment in the ESS during optimization. Since the bound is directly related with the ESS, why not reporting that as well? It is nice to see performance curves that compare other algorithms, but sometimes is even better to have a theory section well integrated with the empirical one. 4- I would remove the offline/online narrative, which is confusing. For example, in the abstract it looks like the algorithm is just optimizing offline, but in reality it does not. Other: - Eq(1): outer integral should be over $\theta$? - line 104: integral again - line 155: "access to set" -> "access to a set" - line 218: swap small o with the plus sign

Reviewer 3



In this work, the authors propose a novel model-free policy search algorithm, namely POIS, which explicitly accounts for uncertainty introduced by the importance sampling by optimizing an objective function that captures the trade-off between the estimated performance improvement and variance injected by the importance sampling. The central idea is to use Rényi divergence and derive the surrogate loss like Equation (9) and Equation (10), which explicitly bound the variance of the importance sampling estimator. One key confusion is the division of the policy optimization methods, I haven’t seen the “control-based” and “parameter-based” categories in the RL literature. Also from the naming of these two categories, it does not mean what the algorithm actually does. Personally, I’d like to view the “parameter-based” method as a Bayesian form parameterization of the policy, whose weights are sampled from some distributions. And they could be merged into a unified form. I am not an expert on Rényi divergence nor the theoretical proof. Maybe these parts could be checked carefully by other reviewers. From the experimental results, it’s a bit unsatisfactory that the performance is almost the same as the previous methods, which seem quite a bit more simpler. I’d like to see the actual benefits from the experimental results, i.e., the benefits of explicit modeling the variance of the importance sampling and whether it actually gets reduced. ==== In the rebuttal, the authors tried to use some better terms to replace "control-based" and "parameter-based". Though the proposed replacement does not seem very satisfactory, the reviewer trusts the authors can find better terms in the final version. And the authors have clarified the confusion in the experiment part.